# Discrete Bayesian Sample Inference for Graph Generation

**Ole Petersen**[*1,4]    **Marcel Kollovieh**[*1,2,3]    **Marten Lienen**[1,3]    **Stephan Günnemann**[1,2,3]

[1] School of Computation, Information and Technology, Technical University of Munich
[2] Munich Center for Machine Learning    [3] Munich Data Science Institute    [4] Listen Labs

Correspondence to: `m.kollovieh@tum.de`

## Abstract

Generating graph-structured data is crucial in applications such as molecular generation, knowledge graphs, and network analysis. However, their discrete, unordered nature makes them difficult for traditional generative models, leading to the rise of discrete diffusion and flow matching models. In this work, we introduce *GraphBSI*, a novel one-shot graph generative model based on Bayesian Sample Inference (BSI). Instead of evolving samples directly, GraphBSI iteratively refines a *belief* over graphs in the continuous space of distribution parameters, naturally handling discrete structures. Further, we state BSI as a stochastic differential equation (SDE) and derive a noise-controlled family of SDEs that preserves the marginal distributions via an approximation of the score function. Our theoretical analysis further reveals the connection to Bayesian Flow Networks and Diffusion models. Finally, in our empirical evaluation, we demonstrate state-of-the-art performance on molecular and synthetic graph generation, outperforming existing one-shot graph generative models on the standard benchmarks Moses and GuacaMol.

## 1 Introduction

Graph structures appear in various domains ranging from molecular chemistry to transportation and social networks. Generating realistic graphs enables simulation of real-world scenarios, augmenting incomplete datasets, and discovering new materials and drugs (Guo & Zhao, 2022; Zhu et al., 2022). However, their unique and complex structure poses challenges to traditional generative models that are designed for continuous data such as images. This has resulted in a diverse landscape of graph generative models, featuring autoregressive models (You et al., 2018) and one-shot models (Kipf & Welling, 2016), including a range of diffusion-based models (Ho et al., 2020).

Recently, Bayesian Flow Networks (BFNs) (Graves et al., 2025) have emerged as a novel class of models that operate on the parameters of a distribution over samples rather than on the samples themselves. This approach is particularly appealing for discrete data, as the parameters of a probability distribution evolve smoothly even when the underlying samples remain discrete. Graph generative models based on BFNs have shown competitive performance in molecule generation (Song et al., 2025). However, operating in parameter space and being motivated through information theory adds a layer of complexity to the BFN framework that hinders its accessibility.

*Bayesian Sample Inference* (BSI) (Lienen et al., 2025) offers a simplified interpretation and generalizes continuous BFNs by viewing generation as a sequence of Bayesian updates that iteratively refine a belief over the unknown sample. The model is trained by optimizing its corresponding ELBO.

This work introduces **GraphBSI**, extending BSI to discrete graphs. Instead of operating on discrete states, GraphBSI evolves on the probability simplex of node and edge categories. We derive BSI for categorical data and show how to generate variably-sized graphs with it. Next, we formulate

---

[*]Equal contribution

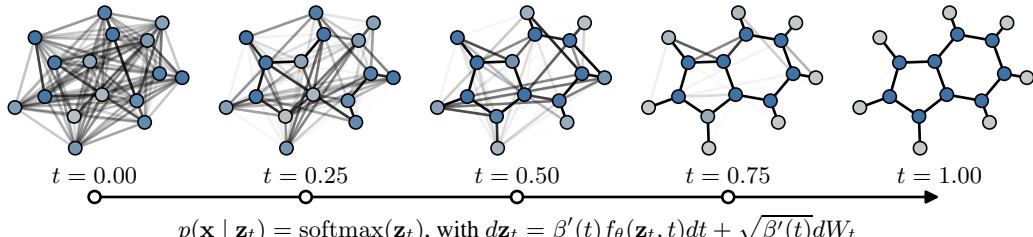

$$p(\mathbf{x} \mid \mathbf{z}_t) = \mathrm{softmax}(\mathbf{z}_t), \text{ with } d\mathbf{z}_t = \beta'(t)f_\theta(\mathbf{z}_t, t)dt + \sqrt{\beta'(t)}dW_t$$

Figure 1: Illustration of GraphBSI's generative process. Nodes and edges are modeled as independent categorical variables. One edge-type is used to represent the non-existence of an edge. The latent variable $\mathbf{z}_t$ represents a *distribution over graphs* rather than a graph itself. The neural network $f_\theta$ smoothly steers this distribution from a random initial distribution $\mathbf{z}_0$ to a distribution concentrated on valid graphs $\mathbf{z}_1$, which is modeled as a Stochastic Differential Equation (SDE).

categorical BSI as an SDE and, via the Fokker–Planck equation, derive a noise-controlled family of SDEs that preserves marginals while interpolating between a deterministic probability-flow ODE and a highly stochastic sampler. Empirically, we demonstrate that GraphBSI achieves state-of-the-art results on the GuacaMol (Brown et al., 2019) and Moses (Polykovskiy et al., 2020) benchmarks for molecule generation. In extensive ablation studies, we show that noise control is a crucial factor for optimizing performance. An overview of our method is shown in Fig. 1.

Our **main contributions** can be summarized as follows:

- We derive BSI for categorical data, enabling, among others, the generation of graphs and sequences. The result generalizes the Bayesian Flow Network (BFN) framework with a simplified interpretation while avoiding limit approximations in the Bayesian update.

- We formulate categorical BSI as an SDE. Through the Fokker-Planck equation, we derive a generalized SDE with a noise-controlling parameter and identical marginals, allowing us to interpolate between a deterministic probability flow ODE and a sampling scheme that overrides all previous predictions with the most recent one.

- We demonstrate that GraphBSI achieves SOTA results across most metrics in the Moses and GuacaMol molecule generation benchmarks with as few as 50 function evaluations, and further gains substantial improvements with 500 function evaluations.

## 2 THE BAYESIAN SAMPLE INFERENCE FRAMEWORK FOR GRAPHS

Bayesian Sample Inference (BSI) (Lienen et al., 2025) is a novel generative modeling framework simplifying and generalizing Bayesian Flow Networks (BFNs) (Graves et al., 2025). While BSI was originally presented for continuous data, we develop a theoretical framework extending BSI to categorical data analogously. We start by introducing the required background knowledge. All proofs are shown in Sec. D.

**Background.** Bayesian Sample Inference (BSI) (Lienen et al., 2025) generates samples by iteratively refining an initial belief $p(\mathbf{x})$ about the sample $\mathbf{x}$ to be generated through noisy measurements $\mathbf{y}$ of $\mathbf{x}$. The initial belief $p(\mathbf{x} \mid \mathbf{z}_0)$ follows a broad isotropic Gaussian with parameters $\mathbf{z}_0 = (\boldsymbol{\mu}_0, \sigma_0)$. The belief is then refined by a sequence of noisy measurements $\mathbf{y}_0, \dots, \mathbf{y}_{k-1}$ that follow Gaussians centered around $\mathbf{x}$. After receiving the measurement $\mathbf{y}_i$, the information contained in it is integrated into our next belief $\mathbf{z}_{i+1}$ through a Bayesian update. Once the belief of $\mathbf{x}$ is sufficiently sharp, we return a sample from it. We train a neural network $f_\theta$ to predict the train sample $\mathbf{x}$ from the information collected about it in the belief $\mathbf{z}_i$ for each timestep $i \in 0, \dots, k-1$. The trained neural network allows us to generate new samples during inference by creating the noisy measurements through an approximation $\hat{\mathbf{x}}_i = f_\theta(\mathbf{z}_i, i)$ of the sample $\mathbf{x}$ in each timestep $i$.

**Extension to categorical data.** Now, we will focus on the case that our data lies on the simplex, i.e., we have a categorical belief for $\mathbf{x}$ over $c$ possible categories, i.e., $x \in \Delta_{c-1}^n \subset [0,1]^{n \times c}$. If we

have access to noisy measurements $\mathbf{y}_i \sim \mathcal{N}(\mathbf{x}, \Sigma^2 = \alpha_i^{-1}I)$ of the sample $\mathbf{x}$, we can infer $\mathbf{x}$ from the measurements using Bayes' theorem in a similar fashion to the continuous case. We start with an initial belief $p(\mathbf{x} \mid \mathbf{z}_0) \sim \text{Cat}(\text{softmax}(\mathbf{z}_0))$, where $\mathbf{z}_0 \in \mathbb{R}^{n \times c}$ are the logits of a categorical distribution with $n$ independent components. Then, we can update the belief parameters $\mathbf{z}$ after observing $\mathbf{y}_i$ using Bayes' theorem.

**Theorem 1.** *Given a prior belief $p(\mathbf{x} \mid \mathbf{z}) = \text{Cat}(\mathbf{x} \mid \text{softmax}(\mathbf{z}))$, after observing $\mathbf{y} \sim \mathcal{N}(\mathbf{y} \mid \mu = \mathbf{x}, \Sigma^2 = \alpha^{-1}\mathbf{I})$ at precision $\alpha$, the posterior belief is $p(\mathbf{x} \mid \mathbf{z}, \mathbf{y}, \alpha) = \text{Cat}(\mathbf{x} \mid \text{softmax}(\mathbf{z}_{\text{post}}))$ with*

$$\mathbf{z}_{\text{post}} = \mathbf{z} + \alpha\mathbf{y}. \tag{1}$$

Now, we can iterate over multiple noisy measurements and update our belief until $p(\mathbf{x} \mid \mathbf{y}_1, \dots, \mathbf{y}_k)$ identifies $\mathbf{x}$ with high probability. Through Theorem 1, we encode the information contained in all these measurements in our updated belief parameters $\mathbf{z}_k$ as $p(\mathbf{x} \mid \mathbf{y}_1, \dots, \mathbf{y}_k) = p(\mathbf{x} \mid \mathbf{z}_k) \sim \text{Cat}(\text{softmax}(\mathbf{z}_k))$ with $\mathbf{z}_k = \mathbf{z}_0 + \sum_i \alpha_i \mathbf{y}_i$.

We process each observation $\mathbf{y}_i$ sequentially, inducing a notion of time. We measure $\mathbf{y}_i$ at time $t_i = \Delta t \cdot i \in [0, 1]$ with $\Delta t = 1/(k+1)$, and the subsequent Bayesian update takes us to $t_{i+1}$. To control the total amount of information added to the belief $p(\mathbf{x} \mid \mathbf{z}_t)$ up to time $t$, we define a monotonically increasing *precision schedule* $\beta \colon [0, 1] \to \mathbb{R}^+$. The measurement $\mathbf{y}_i$ contains the information added in the time interval $[t_i, t_{i+1}]$, and therefore we choose $\alpha_i = \beta(t_{i+1}) - \beta(t_i)$. Note that the update of the logits in Theorem 1 is fundamentally different than that of continuous BSI. Here, the belief components accumulate in each update, whereas in the continuous case, the update is interpolated with its previous state.

**Generative model construction.** We build a generative model for categorical data given the above procedure, similarly as done for BSI with continuous data (Lienen et al., 2025). We begin with a logit $\mathbf{z}_0$ defining the initial belief of the sample $\mathbf{x}$ that we will generate in the end, with $\mathbf{z}_0 \sim \mathcal{N}(\boldsymbol{\mu}_0, \beta_0)$ sampled from a simple prior distribution. As $\mathbf{x}$ is unknown a priori, we cannot measure it, so instead we estimate it from the information we have gathered so far encoded in our latest belief. Let $f_\theta \colon \mathbb{R}^{n \times c} \times [0, 1] \mapsto \Delta_{c-1}^n$ be a neural network with parameters $\theta$ estimating the unknown sample $\mathbf{x}$ behind our observations given our current belief $\mathbf{z}_t$ and time $t$. We estimate $\mathbf{x}$ as $\hat{\mathbf{x}}_i = f_\theta(\mathbf{z}_i, t)$, followed by a noisy measurement $\mathbf{y}_i \sim \mathcal{N}(\hat{\mathbf{x}}_i, \Sigma^2 = \alpha_i^{-1})$ centered around $\hat{\mathbf{x}}_i$ with precision $\alpha_i$. Then, we update our belief with $\mathbf{y}_i$ via Theorem 1. Now, we repeatedly predict $\hat{\mathbf{x}}_i$, measure $\mathbf{y}_i$, and update the belief parameters $\mathbf{z}_{i+1} \leftarrow \mathbf{z}_i + \alpha_i \mathbf{y}_i$ until our belief is sufficiently sharp at $t = 1$. Finally, we return a sample from $\text{Cat}(\mathbf{x} \mid \text{softmax}(\mathbf{z}_1))$. See Alg. 1 for a formal description.

**Evidence Lower Bound.** To train our neural network, we interpret CatBSI as a hierarchical latent variable model to derive an evidence lower bound (ELBO) of the sample likelihood (Kingma & Welling, 2013), providing a natural training target. As latent variables, we choose the beliefs $\mathbf{z}_0, \dots, \mathbf{z}_k$. Their distribution in Alg. 1 factorizes, allowing us to write

$$p(\mathbf{x}) = \mathop{\mathbb{E}}_{p(\mathbf{z}_0) \prod_{i=1}^k p(\mathbf{z}_i \mid \mathbf{z}_{i-1}, \theta)} \left[ p(\mathbf{x} \mid \mathbf{z}_k) \right]. \tag{2}$$

As encoding distribution $q(\mathbf{z}_0, \mathbf{z}_1, \dots, \mathbf{z}_k \mid \mathbf{x})$, we choose the distribution induced under Alg. 1 with a fixed reconstruction $f_\theta(\mathbf{z}, t) = \mathbf{x}$. Thanks to the simple form of Theorem 1, it is straightforward to compute the marginal $q(\mathbf{z}_i \mid \mathbf{x})$:

$$\mathbf{z}_i = \mathbf{z}_0 + \sum_{j=0}^{i-1} \alpha_j \mathbf{y}_j \sim \mathcal{N}(\boldsymbol{\mu}_0 + \beta(t_i)\mathbf{x}, \Sigma^2 = \beta_0 + \beta(t_i)) \tag{3}$$

Equipped with this, we can derive the following ELBO:

**Theorem 2.** *For categorical BSI, the log-likelihood of $\mathbf{x}$ under Alg. 1 is lower bounded by*

$$\log p(\mathbf{x}) \geq \mathop{\mathbb{E}}_{\mathbf{z}_k \sim q(\mathbf{z} \mid \mathbf{x}, t_k)} \left[ \log p(\mathbf{x} \mid \mathbf{z}_k) \right] - \frac{k}{2} \mathop{\mathbb{E}}_{\substack{i \sim \mathcal{U}(0, k-1) \\ \mathbf{z}_i \sim q(\mathbf{z} \mid \mathbf{x}, t_i)}} [(\beta(t_{i+1}) - \beta(t_i)) \| f_\theta(\mathbf{z}_i, t_i) - \mathbf{x} \|_2^2], \tag{4}$$

*where $q(\mathbf{z} \mid \mathbf{x}, t) = \mathcal{N}(\mathbf{z} \mid \boldsymbol{\mu}_0 + \beta(t)\mathbf{x}, \beta_0 + \beta(t)I)$ and $p(\mathbf{x} \mid \mathbf{z}_k) = \text{Cat}(\mathbf{x} \mid \text{softmax}(\mathbf{z}_k))$.*

---

**Algorithm 1** Sampling with Categorical BSI

**Require:** reconstructor $f_\theta$, discretization $k$, precision schedule $\beta : [0,1] \to \mathbb{R}^+$
$\mathbf{z}_0 \sim \mathcal{N}(\boldsymbol{\mu}_0, \beta_0 I)$
**for** $i = 0, \ldots, k-1$ **do**
    $\hat{\mathbf{x}}_i \leftarrow f_\theta(\mathbf{z}_i, t_i)$
    $\alpha_i \leftarrow \beta(t_{i+1}) - \beta(t_i)$
    $\mathbf{y}_i \sim \mathcal{N}(\mu = \hat{\mathbf{x}}_i, \Sigma^2 = \alpha_i^{-1} \cdot I)$
    $\mathbf{z}_{i+1} \leftarrow \mathbf{z}_i + \alpha_i \mathbf{y}_i$
**end for**
$x \sim \text{Cat}(\text{softmax}(\mathbf{z}_k))$
**return** $x$

---

**Algorithm 2** Training Categorical BSI

**while** not converged **do**
    $\mathbf{x} \sim p(\mathbf{x})$
    $\mathbf{z}_0 \sim \mathcal{N}(\boldsymbol{\mu}_0, \beta_0 I)$
    $t \sim \mathcal{U}(0,1)$
    $\alpha = \beta(t) - \beta(0)$
    $\mathbf{y} \sim \mathcal{N}(\mu = \mathbf{x}, \Sigma^2 = 1/\alpha \cdot I)$
    $\mathbf{z} = \mathbf{z}_0 + \alpha \cdot y$
    $\hat{\mathbf{x}} = f_\theta(\mathbf{z}, t)$
    $\mathcal{L} = \beta'(t)/2 \cdot \|\hat{\mathbf{x}} - x\|_2^2$
    $\theta = \theta - \eta \nabla_\theta \mathcal{L}$
**end while**

---

The first term does not depend on $\theta$ and therefore cannot be optimized; we only need to minimize the second term. For $k \to \infty$, we have that $k(\beta(t_{i+1}) - \beta(t_i)) \to \beta'(t_i)$ since $\Delta t = t_{i+1} - t_i = 1/(k+1) \approx 1/k$, and $t_i \sim \mathcal{U}(0,1)$. Maximizing the ELBO for $k \to \infty$ over the dataset above is therefore equivalent to minimizing

$$\mathcal{L} \equiv \mathop{\mathbb{E}}_{\substack{\mathbf{x} \sim p(\mathbf{x}) \\ t \sim \mathcal{U}(0,1) \\ \mathbf{z} \sim q(\mathbf{z}|\mathbf{x},t)}} [\beta'(t)/2 \cdot \|f_\theta(\mathbf{z}, t) - \mathbf{x}\|_2^2] \tag{5}$$

The loss above immediately yields the training procedure Alg. 2. This matches the continuous-time categorical BFN loss up to a constant when $\beta_0 \to 0$, i.e., the prior is a Dirac delta at $t = 0$.

**Adaptation for graphs.** We represent graphs with $N$ nodes as tuples $(X, A)$, where $X \in \Delta_{c_X - 1}^N \subset [0,1]^{N \times c_X}$ are the one-hot encoded categories of each node and $A \in \Delta_{c_A - 1}^{N \times N} \subset [0,1]^{N \times N \times c_A}$ the one-hot encoded categories of each edge, with the first category denoting the absence of an edge. We treat each node and edge as an independent component of the categorical belief, allowing us to apply the categorical BSI framework to graphs. Note that dependence between edges is introduced via our network $f$. We choose a permutation invariant reconstruction network $f_\theta$, resulting in a permutation invariant generative model when the noise is isotropic.

To enable a varying number of nodes in the graph, we first sample a number of nodes $N$ from the marginal node count distribution, and subsequently generate the node and edge values. In practice, this is achieved by masking out inactive nodes and edges for train graphs with fewer nodes.

**Adaptation for sequences** As a general discrete generative model, Categorical BSI is applicable for sequence generation, too. Here, a sequence $S$ of length $l$ with a vocabulary size $v$ is represented in the one-hot-encoded format $S \in \Delta_v^l \subset [0,1]^{l \times v}$. We include an exemplary implementation trained on DNA sequences in Sec. B.

## 3 CATEGORICAL BSI AS A STOCHASTIC DIFFERENTIAL EQUATION

In this section, we analyze the update equation in Theorem 1 and take the infinite-step limit, obtaining an SDE. We then introduce a parameter that controls the stochasticity and yields a family of SDEs with identical marginals.

**SDE Dynamics.** First, we notice that as the number of steps $k$ increases, i.e., $\Delta t := 1/(k+1) \to 0$, the updates in Theorem 1 converge to the following SDE.

**Theorem 3.** *As $\Delta t \to 0$, the update equation in Theorem 1 converges to the following SDE:*

$$d\mathbf{z}_t = \beta'(t) f_\theta(\mathbf{z}_t, t) dt + \sqrt{\beta'(t)} dW_t \tag{6}$$

*where $dW_t$ is a Wiener process and $\mathbf{z}_0 \sim \mathcal{N}(\boldsymbol{\mu}_0, \beta_0 \cdot I)$.*

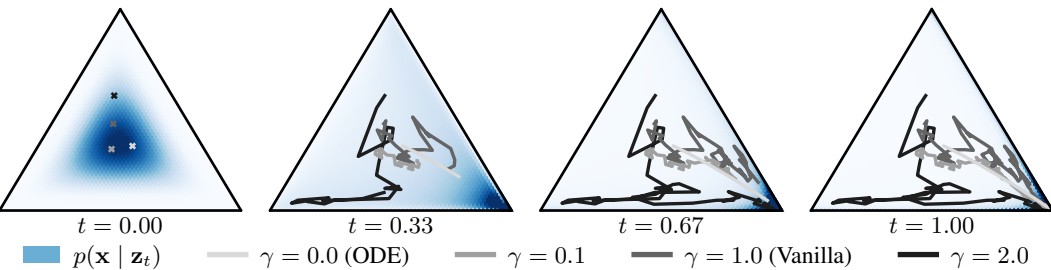

Figure 2: Trajectories of the SDE Theorem 4 for different values of $\gamma$ with three classes and fixed reconstruction $f_\theta(\mathbf{z}_t, t) = \hat{e}_2$. At $\gamma = 0$, the sampler resembles a probability flow ODE as in flow matching. Increasing $\gamma$ leads to noisier trajectories. At $\gamma = 1$, the original SDE in Theorem 3 is recovered, and increasing the noise further makes the trajectories even more volatile. The density function of the marginal distribution $p(\mathbf{x} \mid \mathbf{z}_t)$ (shown in the background) is identical for all $\gamma$.

Note that while the distribution of $\mathbf{z}_0$ is not required to be normal for Theorem 3 itself, it is necessary for the following steps. Phrasing the evolution of the latent variable $\mathbf{z}_t$ as an SDE enables the use of more advanced sampling schemes and allows us to derive a generalized SDE family. The original discrete update in Theorem 1 is recovered by applying an Euler-Maruyama discretization of Eq. (6).

**Generalized SDE.** We now generalize Eq. (6) to a family that preserves the marginal probability paths $p_t(\mathbf{z}_t)$ while controlling stochasticity via the parameter $\gamma$, similar to Karras et al. (2022):

**Theorem 4.** *The SDE in Theorem 3 is generalized by the following family of SDEs with equal marginal densities $p_t(\mathbf{z}_t)$:*

$$d\mathbf{z}_t = \beta'(t)f_\theta(\mathbf{z}_t, t)dt + \frac{\gamma - 1}{2}\beta'(t)\nabla_{\mathbf{z}_t} \log p_t(\mathbf{z}_t)dt + \sqrt{\gamma\beta'(t)}dW_t \tag{7}$$

*where $dW_t$ is a Wiener process and $\mathbf{z}_0 \sim \mathcal{N}(\boldsymbol{\mu}_0, \beta_0 \cdot I)$.*

Setting $\gamma = 0$ yields a deterministic probability flow ODE, equivalent to Xue et al. (2024). Unlike BFNs, however, CatBSI samples the prior belief $p(\mathbf{z} \mid t = 0)$ rather than choosing a fixed prior, naturally avoiding the discontinuity around $t = 0$. Further, choosing $\gamma = 1$ recovers the original SDE in Theorem 3, and larger $\gamma$ produces more stochastic trajectories. We visualize in Figs. 2 and 6 how varying $\gamma$ affects the dynamics for toy examples. Although the marginal distributions are equal for all $\gamma$ in theory, the empirical performance varies as $\nabla_{\mathbf{z}_t} \log p_t(\mathbf{z}_t)$ is not available in closed form. Higher stochasticity allows the model to correct errors made in previous sampling steps but requires a finer discretization (see Sec. 4.3). In the limit $\gamma \to \infty$, the sampler effectively overwrites the current state completely in every step (see Sec. C.3). To turn Eq. (7) into a practical sampling algorithm, we approximate the score function $\nabla_{\mathbf{z}_t} \log p_t(\mathbf{z}_t)$, as described in the following.

**Theorem 5.** *The BSI loss Eq. (5) also is a score matching loss with the score model $s_\theta(\mathbf{z}, t)$ parameterized as*

$$s_\theta(\mathbf{z}, t) \equiv \frac{\boldsymbol{\mu}_0 + \beta(t)f_\theta(\mathbf{z}, t) - \mathbf{z}}{\beta(t) + \beta_0} \stackrel{!}{\approx} \nabla_{\mathbf{z}} \log p_t(\mathbf{z}) \tag{8}$$

**Discretization and integration.** As the SDE is not solvable in closed form, we resort to numerical sampling. While a simple Euler-Maruyama (EM) approach performs well on sufficiently fine time grids, we find that integrating a locally linearized SDE within each step can improve sample quality for low numbers of neural function evaluations (see Sec. 4.3). More specifically, we freeze the reconstructor $\hat{\mathbf{x}} = f_\theta(\mathbf{z}_t, t)$ over the time interval $[t, t + \Delta t]$, representing an Ornstein-Uhlenbeck process. This allows us to solve the SDE analytically within this interval.

**Theorem 6.** *Fixing the prediction $\hat{\mathbf{x}} = f_\theta(\mathbf{z}_t, t)$ and the values $\beta = \beta(t + \Delta t/2)$, $\beta' = \beta'(t + \Delta t/2)$ in Eq. (7) in a time interval $[t, t + \Delta t]$ yields an Ornstein-Uhlenbeck (OU) process with the exact marginal*

$$\mathbf{z}_{t+\Delta t} \sim m + (\mathbf{z}_t - m)e^{-\kappa\Delta t} + \sqrt{\frac{\gamma\beta'}{2\kappa}(1 - e^{-2\kappa\Delta t})} \cdot \mathcal{N}(0, I), \tag{9}$$

**Algorithm 3** Euler-Maruyama Sampling

**Require:** reconstructor $f_\theta$, discretization $\Delta t$, precision schedule $\beta : [0, 1] \to \mathbb{R}^+, \gamma \geq 0$
$\quad \mathbf{z} \sim \mathcal{N}(\boldsymbol{\mu}_0, \beta_0 I)$
$\quad$ **for** $t = 0, \Delta t, 2\Delta t, \ldots, 1 - \Delta t$ **do**
$\quad\quad \hat{\mathbf{x}} \leftarrow f_\theta(\mathbf{z}, t)$
$\quad\quad \mathbf{s}_\theta \leftarrow \frac{\boldsymbol{\mu}_0 + \beta(t)\hat{\mathbf{x}} - \mathbf{z}}{\beta(t) + \beta_0}$
$\quad\quad \boldsymbol{\mu} \leftarrow \beta'(t)(\hat{\mathbf{x}} + \frac{\gamma - 1}{2}\mathbf{s}_\theta)$
$\quad\quad \sigma \leftarrow \sqrt{\gamma \beta'(t)}$
$\quad\quad \mathbf{z} \leftarrow \mathbf{z} + \boldsymbol{\mu}\Delta t + \sigma\sqrt{\Delta t} \cdot \mathcal{N}(0, I)$
$\quad$ **end for**
$\quad$ **return** $\text{Quantize}(f_\theta(\mathbf{z}, t = 1))$

**Algorithm 4** Ornstein-Uhlenbeck Sampling

**Require:** reconstructor $f_\theta$, discretization $\Delta t$, precision schedule $\beta : [0, 1] \to \mathbb{R}^+, \gamma > 1$
$\quad \mathbf{z} \sim \mathcal{N}(\boldsymbol{\mu}_0, \beta_0 I)$
$\quad$ **for** $t = \Delta t/2, \Delta t + \Delta t/2, \ldots, 1 - \Delta t/2$ **do**
$\quad\quad \hat{\mathbf{x}} \leftarrow f_\theta(\mathbf{z}, t)$
$\quad\quad \kappa \leftarrow \frac{(\gamma - 1)\beta'(t)}{2(\beta_0 + \beta(t))}$
$\quad\quad m \leftarrow \boldsymbol{\mu}_0 + (\beta(t) + \beta'(t)/\kappa)\hat{\mathbf{x}}$
$\quad\quad \sigma^2 \leftarrow \frac{\gamma \beta'(t)}{2\kappa}(1 - e^{-2\kappa\Delta t})$
$\quad\quad \mathbf{z} \leftarrow m + (\mathbf{z} - m)e^{-\kappa\Delta t} + \sqrt{\sigma^2} \cdot \mathcal{N}(0, I)$
$\quad$ **end for**
$\quad$ **return** $\text{Quantize}(f_\theta(\mathbf{z}, t = 1))$

*where* $\kappa = \frac{(\gamma - 1)\beta'}{2(\beta_0 + \beta)}$, $m = \boldsymbol{\mu}_0 + (\beta + \beta'/\kappa)\hat{\mathbf{x}}$.

Note that the OU discretization converges towards the EM scheme for $\Delta t \to 0$ (see Sec. C.1).

**Quantizing instead of sampling.** If the belief precision at $t = 1$ is sufficiently sharp, the final sampling step in Alg. 1 is de facto deterministic. However, this presents an opportunity to improve sampling efficiency: In the last few steps, simply sampling from the belief would yield too noisy samples, but the belief contains enough information so that the reconstructor can make a perfect reconstruction of it (see Fig. 5). Therefore, we can instead stop at a lower final precision and return reconstruction projected on the sample space through a quantization. Employing the discretization schemes yields Algs. 3 and 4.

We also allow a nonuniform time grid. Following Karras et al. (2022), we introduce a parameter $\rho$ that controls the distribution of function evaluations over the time grid:

$$t_i = \left(\frac{i}{k}\right)^\rho, \qquad i = 0, 1, \ldots, k. \tag{10}$$

Here, $\rho = 1$ recovers a uniform grid; larger $\rho$ concentrates steps near the beginning ($t \approx 0$), whereas smaller $\rho$ concentrates them near the end ($t \approx 1$).

# 4 EXPERIMENTS

In this section, we present our empirical results. We benchmark our model against state-of-the-art baselines from the diffusion and flow-matching literature on unconditional molecular and synthetic graph generation. The GuacaMol and Moses benchmarks for molecular generation (Brown et al., 2019; Polykovskiy et al., 2020) serve as our primary evaluation datasets. Additionally, we conduct ablation studies to analyze the impact of various components and hyperparameters on the model's performance. Further, we report results on the synthetic planar, tree, and stochastic block model graph generation tasks (Bergmeister et al., 2024; Martinkus et al., 2022).

## 4.1 EXPERIMENTAL SETUP

**Datasets.** To test performance on real-world graphs, we train GraphBSI on the Moses (Polykovskiy et al., 2020) and GuacaMol (Brown et al., 2019) datasets for molecular generation. We extract graphs out of the dataset smiles with RDKit RDKit (2025) and construct the node features $X$ and adjacency matrix $A$ in the format described in Sec. 2, where atom- and bond types correspond to node- and edge categories, respectively. Further, we include results for the planar, tree, and stochastic block model (Martinkus et al., 2022; Bergmeister et al., 2024) synthetic graph generation datasets. Find a summary in Tab. 8.

**Evaluation metrics.** We follow the standard evaluation practices as established by Polykovskiy et al. (2020); Brown et al. (2019); Preuer et al. (2018) for molecule generation and Martinkus et al.

(2022); Bergmeister et al. (2024) for synthetic graph generation. Find a detailed description in Tabs. 9 and 10.

**Practical considerations.** The reconstruction network $f_\theta$ is parameterized using the same graph transformer architecture as Qin et al. (2025); Vignac et al. (2023), with the node- and edge logits and class probabilities, entropy, random walk features, and sinusoidal embeddings (Vaswani et al., 2017) of the timestep $t$ with frequencies proposed by Lienen et al. (2024) as features. Empirically, we find that an exponential precision schedule with a final precision that allows for a near-perfect reconstruction maximizes performance (see Tab. 7 and Figs. 5 and 7). For both latent node- and edge classes, we choose a normal prior with the marginal distribution over the dataset and a small variance of $1.0$. Finally, we apply a preconditioning scheme where the neural network predicts the difference between the belief and the true sample, setting $f_\theta(z, t) = \mathrm{softmax}(z + \tilde{f}_\theta(z, t))$.

**Evaluation** After training to convergence, we evaluate the benchmark metrics for both discretization schemes Algs. 3 and 4. For both molecule generation benchmarks, we report results with a compute budget of 50 and 500 discretization steps. In each of the four configurations (2 discretization schemes, 2 numbers of steps), we optimize the noise level $\gamma$ and report the best result. Find the final configurations in Tab. 7. For the synthetic graph generation benchmarks, we report results with the best-performing noise level and the Ornstein-Uhlenbeck discretization with 1000 function evaluations.

## 4.2 RESULTS

**Molecule Generation.** As illustrated in Tab. 1, GraphBSI is competitive with 50 steps with both discretization schemes for both molecule benchmarks, achieving state-of-the-art results on the majority of the metrics. Notably, GraphBSI outperforms DeFoG with both discretization schemes on all metrics except novelty on Moses. On most metrics, the OU discretization performs better than the EM scheme. At the full 500 steps, GraphBSI with the OU discretization outperforms all existing models on all metrics on GuacaMol, saturating validity and consistently exceeding the state-of-the-art. The EM scheme performs slightly worse than OU on most metrics, but remarkably surpasses the state-of-the-art on the FCD metric, reducing it from 1.07 to 0.72 on Moses. Find an extended comparison in Tab. 5.

Table 1: Results on the GuacaMol and Moses benchmarks for molecular generation with 50 and 500 sampling steps and the Euler-Maruyama (EM) and Ornstein-Uhlenbeck (OU) discretization.

| Model | Steps | GuacaMol | | | | | Moses | | | | | | |
|---|---|---|---|---|---|---|---|---|---|---|---|---|---|
| | | Val. ↑ | V.U. ↑ | V.U.N. ↑ | KL ↑ | FCD ↑ | Val. ↑ | Uniq. ↑ | Nov. ↑ | Filters ↑ | FCD ↓ | SNN ↑ | Scaf ↑ |
| Train Set | | 100.0 | 100.0 | 0.0 | 99.9 | 92.8 | 100.0 | 100.0 | 0.0 | 100.0 | 0.01 | 0.64 | 99.1 |
| DeFoG | 50 | 91.7 | 91.7 | 91.2 | 92.3 | 57.9 | 83.9 | 99.9 | **96.9** | 96.5 | 1.87 | 0.50 | **23.5** |
| GraphBSI (EM) | 50 | 97.5 | 97.5 | 97.2 | 90.7 | 65.6 | 99.3 | 100.0 | 96.5 | 96.9 | **1.06** | 0.50 | 15.2 |
| GraphBSI (OU) | 50 | **99.2** | **99.2** | **98.7** | **93.7** | **71.3** | **99.7** | 100.0 | 94.6 | **98.2** | 1.19 | **0.52** | 15.1 |
| DiGress | 500 | 85.2 | 85.2 | 85.1 | 92.9 | 68.0 | 85.7 | 100.0 | 95.0 | 97.1 | 1.19 | 0.52 | 14.8 |
| DisCo | 500 | 86.6 | 86.6 | 86.5 | 92.6 | 59.7 | 88.3 | **100.0** | **97.7** | 95.6 | 1.44 | 0.50 | 15.1 |
| Cometh | 500 | 98.9 | 98.9 | 97.6 | 96.7 | 72.7 | 90.5 | 99.9 | 92.6 | 99.1 | 1.27 | 0.54 | **16.0** |
| DeFoG | 500 | 99.0 | 99.0 | 97.9 | 97.7 | 73.8 | 92.8 | 99.9 | 92.1 | 98.9 | 1.95 | 0.55 | 14.4 |
| GraphBFN | 500 | - | - | - | - | - | 98.5 | 99.8 | 89.0 | 98.3 | 1.07 | **0.59** | 10.0 |
| GraphBSI (EM) | 500 | 98.8 | 98.8 | **98.3** | 94.6 | **82.6** | 99.8 | 100.0 | 92.5 | 99.1 | **0.72** | 0.54 | 14.3 |
| GraphBSI (OU) | 500 | **99.6** | **99.6** | 98.2 | **98.4** | 80.3 | **99.9** | 100.0 | 90.7 | **99.2** | 0.90 | 0.55 | 12.7 |

**Synthetic Benchmarks.** As shown in Tab. 2, GraphBSI achieves competitive results on the synthetic graph generation benchmarks. Our model saturates validity on the planar- and tree graph generation tasks, and achieves adequate validity on the stochastic block model graphs. The mean ratio as a measure of distribution similarity is competitive on all three datasets, even though the metric should be taken with a grain of salt due to the small dataset size of only 128 graphs, resulting in high uncertainty in the evaluation.

Table 2: Results on the synthetic graph generation benchmarks. Like DeFoG, we generate 40 graphs five times and report the mean and standard deviation over the runs.

| Model | Steps | Planar | | Tree | | SBM | |
|---|---|---|---|---|---|---|---|
| | | V.U. ↑ | Ratio ↓ | V.U. ↑ | Ratio ↓ | V.U. ↑ | Ratio ↓ |
| Train Set | | 100.0 | 1.0 | 100.0 | 1.0 | 85.9 | 1.0 |
| DeFoG | 50 | **95.0 ± 3.2** | **3.2 ± 1.1** | 73.5 ± 9.0 | 2.5 ± 1.0 | **86.5 ± 5.3** | **2.2 ± 0.3** |
| GraphBSI (EM) | 50 | 7.5 ± 1.0 | 47.5 ± 4.3 | 89.0 ± 7.0 | **2.1 ± 0.8** | 61.5 ± 5.8 | 4.2 ± 1.4 |
| GraphBSI (OU) | 50 | 38.5 ± 8.6 | 18.0 ± 3.2 | **96.0 ± 1.2** | 2.5 ± 0.9 | 53.0 ± 7.5 | 51.4 ± 4.0 |
| HSpectre | 256 | 95.0 | 2.1 | **100.0** | 4.0 | 75.0 | 10.5 |
| DiGress | 1000 | 77.5 | 5.1 | 90.0 | 1.6 | 60.0 | **1.7** |
| DeFoG | 1000 | 99.5 ± 1.0 | **1.6 ± 1.0** | 96.5 ± 2.6 | 1.6 ± 0.4 | 90.0 ± 5.1 | 4.9 ± 1.3 |
| GraphBFN | 1000 | 96.7 | - | - | - | 87.5 | - |
| GraphBSI (EM) | 1000 | **100.0 ± 0.0** | 3.8 ± 1.0 | 96.5 ± 3.7 | **1.3 ± 0.4** | 50.5 ± 4.6 | 11.3 ± 1.4 |
| GraphBSI (OU) | 1000 | **100.0 ± 0.0** | 3.2 ± 0.6 | **100.0 ± 0.0** | 1.8 ± 0.5 | 77.5 ± 2.7 | 4.6 ± 1.1 |

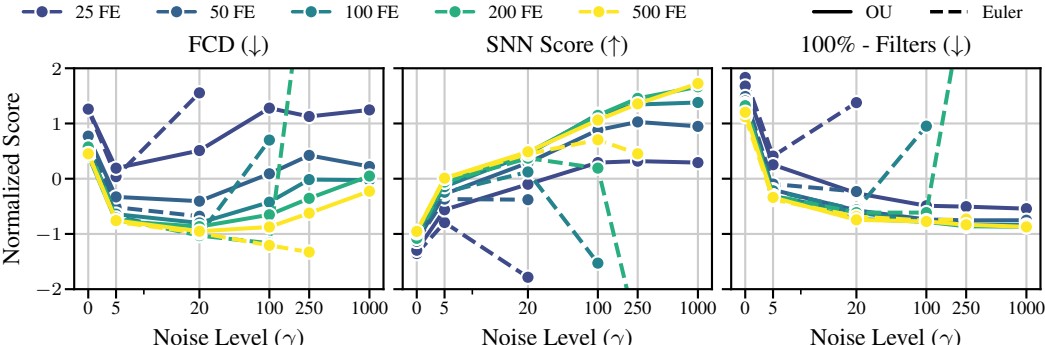

Figure 3: Normalized metrics (zero mean, unit variance) vs. noise level $\gamma$ for different numbers of function evaluations (FE) and discretization schemes. Our custom Ornstein-Uhlenbeck discretization scheme is denoted as OU, while the standard Euler-Maruyama scheme is written as Euler. Some values for the Euler scheme are missing since the sampler becomes unstable if $\gamma \cdot \Delta t$ becomes too large (see Sec. C.2).

## 4.3 ABLATION STUDIES

**Noise level.** To test the effect of the compute budget, noise level, and discretization scheme on performance, we conduct a grid search over the number of function evaluations (NFEs) in $\{25, 50, 100, 200, 500\}$, noise levels $\gamma$ in $\{0.0, 5.0, 20.0, 100.0, 250.0, 1000.0\}$, and both discretization schemes on the Moses dataset. As shown in Fig. 3, performance in both discretization schemes is closely related at low noise levels, which is to be expected since both discretize the same SDE. Higher compute budgets lead to better performance. However, the Euler-Maruyama scheme becomes unstable at higher noise levels, leading to a significant drop in performance (see Sec. C.2). In contrast, the Ornstein-Uhlenbeck scheme remains stable, and both the SNN score and Filters metric benefit from higher noise levels. The FCD metric is optimal at a medium noise level between 20 and 100. With a few exceptions, the Ornstein-Uhlenbeck scheme matches or outperforms the Euler-Maruyama scheme at all compute budgets and noise levels. Novelty suffers from increased noise levels and compute budgets, which is consistent with the model generating samples closer to the training data distribution. Notably, all metrics perform poorly at a noise level of 0.0, which corresponds to the probability flow ODE (equivalent to Xue et al. (2024)). Fig. 8 illustrates that optimizing the noise level is a key driver in the performance gains of our model.

**Non-uniform timesteps.** To test whether a fine discretization is more important at some timesteps compared to others, we analyze the effect of non-uniform timestepping, putting a finer discretization at either earlier or later timesteps. As shown in Fig. 4, SNN and Moses Filters remain mostly unaffected by the choice of $\rho$; only the FCD displays a clear trend. A finer discretization at later timesteps ($\rho < 1$) improves the FCD at 50 function evaluations in both discretization schemes and at 500 evaluations in the Ornstein-Uhlenbeck scheme.

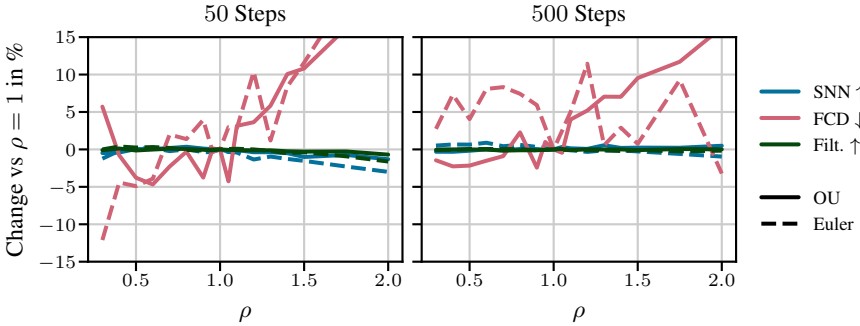

Figure 4: Performance change for changes in the non-uniform timestepping parameter $\rho$ in $t_i = (i/k)^\rho$ for $i = 0, 1, \ldots, k$ compared to the uniform case $\rho = 1$. $\rho < 1$ results in a finer discretization at later timesteps, while $\rho > 1$ corresponds to finer discretization at earlier steps.

**Precision schedule.** We find that while an exponential precision schedule yields the best results, the difference compared to a simple linear schedule is negligible (see Tab. 5). One parameter that significantly affects performance is the final precision $\beta(t = 1)$. As illustrated in Fig. 7, an excessively large final precision wastes sampler iterations in the final steps, and a too small final precision results in noisy samples. Ideally, the reconstructor is just able to predict the train samples flawlessly at $\beta(t = 1)$. Finally, we isolate the effect of sampling the belief at $t = 0$ instead of taking a fixed value, as with BFNs, by training a new model with a smaller initial variance of $\beta_0 = 0.05$, compared to the standard $\beta_0 = 1.0$. Tab. 5 shows that for both values of $\beta_0$, the OU sampler outperforms the Flowback (Song et al., 2025) sampler on most metrics. Surprisingly, the performance of the Flowback sampler drops significantly when $\beta_0$ is increased, while a higher value of $\beta_0$ improves performance for the OU sampler.

We conclude that two key factors are crucial for the performance gains of GraphBSI: First, the noise control, and second, a final precision that is just high enough for a perfect reconstruction. The exact precision schedule and non-uniform time-stepping show only a marginal contribution.

## 5    RELATED WORK

Graph generation presents three main challenges compared to image and text generation: (1) graphs are discrete structures, unlike images, which are continuous; (2) graphs have a variable shape, with both the number and arrangement of nodes and edges changing across samples, unlike the fixed dimensions of images; and (3) nodes in graphs lack a natural order, in contrast to text, where tokens follow a well-defined sequence. Various approaches have been proposed to tackle these challenges.

**Autoregressive models** have proven successful in text generation by sequentially predicting the next token based on previous ones (Brown et al., 2020). Applied to graphs, these models generate nodes and edges one by one, maintaining the graph structure as they proceed. This approach has been used for tasks such as molecule and social-network generation (You et al., 2018; Liao et al., 2020). However, autoregressive models violate permutation invariance by relying on a specific node ordering.

**One-shot models** address the ordering challenge by generating the entire graph in a single step, without relying on a specific node ordering. Examples include Variational Autoencoders (Kingma & Welling, 2013), GANs (Cao & Kipf, 2022), normalizing flows (Liu et al., 2019), and discrete flow matching (Gat et al., 2024; Qin et al., 2025).

**Diffusion models** have emerged as a powerful class of one-shot generative models for continuous data such as images (Sohl-Dickstein et al., 2015; Ho et al., 2020). Their core idea is to learn a generative process that gradually transforms noise into clean data by reversing a diffusion process with a neural network. Noise is typically applied independently to each pixel in images or to each node in graphs, naturally resulting in a permutation-invariant model when combined with a Graph Neural Network (GNN) (Niu et al., 2020). A variable number of nodes can be handled by conditioning the diffusion process on the node count, e.g., by first sampling a node mask and then applying diffusion to the masked graph (Niu et al., 2020; Qin et al., 2025). To improve scalability, hybrid methods that

reverse a coarsening process and generate local structures with a diffusion model have also been proposed (Bergmeister et al., 2024).

**Discrete diffusion** addresses the discreteness of graphs. The most straightforward approach relaxes discrete data to a continuous space, applies diffusion, and quantizes the generated outputs back to the discrete space in a final step (Niu et al., 2020; Jo et al., 2022; 2024). Alternatively, one can use discrete diffusion in which the state is perturbed via a Markovian transition matrix in discrete time steps (often including an absorbing state) (Austin et al., 2023); this has been applied to graphs (Vignac et al., 2023; Haefeli et al., 2023). A related recent approach uses a continuous-time Markov chain for the discrete diffusion process (see (Campbell et al., 2022)), which allows more flexible sampling on graphs (Siraudin et al., 2024; Xu et al., 2024).

**Bayesian Flow Networks** Graves et al. (2025) propose a conceptually distinct approach to discrete generative models: diffusion is applied to the *parameters of a distribution over samples* rather than to the samples themselves. BFNs can be interpreted as an SDE, enabling more efficient sampling algorithms (Xue et al., 2024). This provides a solid theoretical foundation for diffusion on discrete data while retaining the benefits of smooth parameter changes, and it achieves competitive performance on protein and graph generation (Atkinson et al., 2025; Song et al., 2025; Tao & Abe, 2025). The flexible design of BFNs also permits joint generation of continuous and discrete quantities, for example the 3D positions, atom types, and charges in molecular generation (Song et al., 2024).

**Bayesian Sample Inference** Lienen et al. (2025) extends BFNs by adding a prior over the distribution parameters and offers a simplified interpretation for the continuous-data case. Kollovieh et al. (2025) used the BSI framework to derive their generative model for hierarchies. However, they do not generalize the framework, i.e., do not derive SDE-based sampling algorithms, and do not optimize an ELBO as they specifically focus on hierarchy generation.

## 6 CONCLUSION

In this work, we introduce **GraphBSI**, a novel generative model for graphs based on Bayesian Sample Inference with state-of-the-art performance in large molecule generation benchmarks. Similar to Bayesian Flow Networks, GraphBSI iteratively refines a belief over the graph structure, modeled as a categorical distribution over adjacency matrices, through Bayesian updates. We show that in the limit of infinitesimal time steps, GraphBSI converges to a Stochastic Differential Equation (SDE). Further, we employ the Fokker-Planck equation to derive a generalized SDE with a tunable noise parameter, allowing us to interpolate between a deterministic probability flow ODE, the original SDE, and a substantially more volatile sampler. We demonstrate that GraphBSI achieves state-of-the-art performance on the GuacaMol and Moses benchmarks for large molecule generation, outperforming existing models on nearly all metrics. Finally, in our ablations we empirically show that noise control critically influences performance.

**Limitations and Future Work.** GraphBSI, in its current implementation, suffers from the quadratic scaling of compute and memory requirements in the number of nodes that comes with the application of a graph transformer. Exploring a more memory-efficient graph neural network architecture to generate larger graphs would be a promising avenue for future research. Further, while GraphBSI allows for variable-sized graphs, the number of nodes is sampled beforehand instead of jointly generated with the graph features. Allowing for nodes to appear or disappear while generating the graph, similar to jump diffusion (Campbell et al., 2023), might result in a more flexible generative process.

## ACKNOWLEDGMENTS

This paper is supported by the DAAD programme Konrad Zuse Schools of Excellence in Artificial Intelligence, sponsored by the Federal Ministry of Research, Technology and Space."

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

## A    RELATIONSHIP TO BFNS AND DIFFUSION MODELS

### A.1    RELATIONSHIP TO BFNS

There is a close equivalence between Categorical Bayesian Sample Inference (BSI) and Categorical Bayesian Flow Networks (BFNs). In fact, Categorical BFNs can be seen as a special case of Categorical BSI with a specific choice of prior distribution and noise schedule. The dynamics of BFNs are recovered when choosing the sampler in Eq. (7) with $\gamma = 1$ and $\beta_0 = 0$ to parametrize $\mathbf{z}_0$, i.e., making the prior logits deterministic. Note that we require $\beta_0 > 0$ to avoid numerical issues when approximating the score function. This generalized SDE allows BSI to vary stochasticity. Intuitively, increasing stochasticity allows the model to overwrite errors from previous predictions (see Sec. C.3 for a discussion on the extreme case), and empirically, increasing stochasticity proves crucial for performance Fig. 3. To illustrate this, we will show the relationship between the components of both frameworks.

**Input Distribution** Both BFNs and categorical BSI parameterize the distribution over the data $\mathbf{x}$ using a categorical distribution. The logits are denoted as $\mathbf{z}$ in BSI and as $\theta$ in BFNs. In BSI, the parameters $\mathbf{z}$ are the logits of a categorical distribution, i.e., $p(\mathbf{x} \mid \mathbf{z}) \sim \mathrm{Cat}(\mathrm{softmax}(\mathbf{z}))$. In BFNs, the parameters $\theta$ are the probabilities of each category, i.e., $p(\mathbf{x} \mid \theta) \sim \mathrm{Cat}(\theta)$. The two parameterizations are equivalent since $\theta = \mathrm{softmax}(\mathbf{z})$ and $\mathbf{z} = \log(\theta)$ (up to an additive constant).

**Output Distribution** The output distribution in BFNs is an intermediate distribution that is not needed in BSI.

**Prior Distribution** While Categorical BSI includes a normal prior distribution over the logits of the categorical distribution ($p(\mathbf{z} \mid t = 0) \sim \mathcal{N}(\boldsymbol{\mu}_0, \beta_0 I)$), Categorical BFNs fix the parameters to $\theta_0 = 1/K$. Therefore, categorical BFNs can be seen as a special case of categorical BSI with $\boldsymbol{\mu}_0 = 0$ and $\beta_0 = 0$.

**Sender Distribution** The sender distribution in categorical BFNs is an intermediate distribution that is not required in categorical BSI.

**Receiver Distribution** The sender distribution in categorical BFNs is given as

$$p_R(\mathbf{y} \mid \mathbf{x}, \alpha) \sim \sum_k \mathrm{softmax}(\Psi(\theta))_k \mathcal{N}(\alpha(K\hat{e}_k - 1), \alpha K I)$$

It corresponds to the noisy measurement distribution in categorical BSI, $p(\mathbf{y} \mid \mathbf{x}, \alpha) \sim \mathcal{N}(\hat{\mathbf{x}}, 1/\alpha I)$. Note that for $\alpha \to 0$, it holds that:

$$p_R(\mathbf{y} \mid \mathbf{x}, \alpha) \sim \mathcal{N}(\alpha(K\mathrm{softmax}(\Psi(\theta)) - 1), \alpha K I)$$

The sender distribution for $\alpha \to 0$ is an affine transformation of the noisy observation function for BSI: If we set $\mathbf{y} \sim p(\mathbf{y} \mid \mathbf{x}, \alpha) = \mathcal{N}(\hat{\mathbf{x}}, 1/\alpha I)$ and compute $y' = \alpha(K\mathbf{y} - 1)$, then $y' \sim p_s(y' \mid \mathbf{x}, \alpha)$, where $\mathrm{softmax}(\Psi(\theta))$ corresponds to the sample reconstruction $\hat{\mathbf{x}}$. Thus, in the small-$\alpha$-limit, the two distributions have same-order approximation and therefore contain the same information. However, in the formulation of categorical BSI, we can directly see that $\mathbf{y}$ is a noisy observation of $\mathbf{x}$ and we do not require computing the distribution as a limit of a multinomial distribution as in BFNs.

**Bayesian Update Function** The Bayesian update function in categorical BFNs (Graves et al., 2025, Eq. 171) is the equivalent of Theorem 1 in categorical BSI. The update is simplified for BSI since the belief parameters are in logit space instead of probability space. Furthermore, the scaling of the receiver distribution leads to an extra factor of $\alpha$ in categorical BSI.

**Bayesian Update Distribution** This is an intermediate that is not required in categorical BSI.

**Accuracy Schedule** The accuracy schedule can be chosen freely in categorical BSI. In categorical BFNs, the accuracy schedule is chosen as $\beta(t) = t^2 \beta(1)$.

**Bayesian Flow Distribution** The Bayesian flow distribution in categorical BFNs corresponds to Eq. (3) in categorical BSI. The two distributions are equivalent up to an affine transformation of the variable, as explained above.

**Continuous Time Loss** The continuous time loss in categorical BFNs (Graves et al., 2025, Eq. 205) corresponds to Eq. (5) in categorical BSI. Both are the L2 loss between the reconstruction and the one-hot encoded data.

**SDE formulation** Both BSI and BFN sampling can be formulated as SDEs. Here, Theorem 3 corresponds to (Xue et al., 2024, Eq. 24). To do so, the authors also operate on the logits of the categorical distribution instead of the probabilities.

**Score function approximation** The score function approximation for categorical BFNs (Xue et al., 2024, Eq. 28) corresponds to Theorem 5 for $\beta_0 = 0$ up to a constant. Note that a value of $\beta_0 > 0$ avoids the division by zero in the score function approximation at $t = 0$.

## A.2 RELATIONSHIP TO DIFFUSION MODELS.

The logits $\mathbf{z}$ evolve in a way that closely resembles a diffusion process in logit space. From Theorem 1 we have our denoising dynamics

$$p(\mathbf{z}_{t+1} \mid \mathbf{z}_t, \mathbf{x}) = \mathcal{N}(\mathbf{z}_t + \alpha_t \mathbf{x}, \; \alpha_t \mathbf{I}). \tag{11}$$

Moreover, the marginal of $\mathbf{z}_t$ is given by

$$p(\mathbf{z}_t \mid \mathbf{x}) = \mathcal{N}\big(\mu_0 + \beta(t)\mathbf{x}, \; (\beta_0 + \beta(t))\mathbf{I}\big) \tag{12}$$

(see Eq. (3)). We define the corresponding "noising" process as the reverse-time conditional $p(\mathbf{z}_t \mid \mathbf{z}_{t+1}, \mathbf{x})$. Using the standard Gaussian conditioning formula (Murphy, 2012, Eq. 4.125), we obtain

$$p(\mathbf{z}_t \mid \mathbf{z}_{t+1}, \mathbf{x}) = \mathcal{N}\left( \frac{(\beta_0 + \beta(t))\mathbf{z}_{t+1} + \alpha_t \mu_0 - \alpha_t \beta_0 \mathbf{x}}{\beta_0 + \beta(t) + \alpha_t}, \; \frac{\alpha_t(\beta_0 + \beta(t))}{\beta_0 + \beta(t) + \alpha_t} \mathbf{I} \right). \tag{13}$$

Thus, the reverse transition is Gaussian, analogous to the posterior $q(\mathbf{x}_{t-1} \mid \mathbf{x}_t, \mathbf{x}_0)$ in standard diffusion models. While this is not a typical diffusion process in the sense that the derived forward dynamics over $\mathbf{z}_t$ are generally non-Markovian, related non-Markovian formulations have been proposed before (Song et al., 2020). Interestingly, a Markovian process is recovered when setting $\beta_0 = 0$, which coincides with the original BFN parameterization (Graves et al., 2025).

## A.3 RELATIONSHIP WITH FLOW MATCHING MODELS

At noise level $\gamma = 0$, Categorical BSI is closely related to Flow Matching. The sampling SDE Eq. (7) becomes an ODE where the right-hand side can be interpreted as an approximation of the flow field to follow. However, we do not train to directly predict the flow field, but to reconstruct the clean sample. Similar to Dirichlet Flow Matching (DFM), Stark et al. (2024), Categorical BSI operates on a distribution over the simplex. However, while Categorical BSI uses the logits of a categorical distribution as a latent variable, DFM employs a mixture of Dirichlet distributions.

## B BSI FOR SEQUENCE GENERATION

Categorical BSI can generate general categorical data - it is not restricted to graphs. In this section, we demonstrate this capability empirically by training a categorical BSI model to generate sequences. We represent sequences with length $l$ and a vocabulary $v$ in the one-hot encoded format as $S \in \Delta_v^l \subset [0,1]^{l \times v}$. We call the resulting model SeqBSI.

Employing the same reconstruction model as Stark et al., 2024; Davis et al., 2024, a Convolutional Neural Network. We train on the toy dataset from Davis et al. (2024) with $l = 4$ and $v \in \{5, 10, 20, 40, 60, 80, 100, 120, 140, 160\}$ as well as a dataset of enhancer DNA sequences from fly brain cells Janssens et al. (2022) with $l = 500$ and $v = 4$ nucleotide bases.

Following Stark et al. (2024), we report the KL divergence for the toy task and the Fréchet Biological Distance (FBD) as a measure of distribution similarity. As demonstrated in Tab. 3, SeqBSI slightly outperforms Dirichlet Flow Matching (Stark et al., 2024) in the flybrain task. The comparison with Fisher Flow Matching on this metric is difficult, as their evaluation shows vastly different results for Dirichlet flow matching than the results reported in their own paper. On the toy dataset task, SeqBSI outperforms Dirichlet Flow Matching and is competitive with Fisher Flow Matching (see Fig. 9).

Table 3: Results on the enhancer DNA sequence dataset

| Model | Steps | flybrain | |
|---|---|---|---|
| | | FBD ↓ | |
| Random Sequence | | 876.0 | |
| Language Model | 500 | 25.2 | |
| Linear FM | 100 | 15.0 | |
| Dirichlet FM | 100 | 15.2 | |
| SeqBSI (OU) | 100 | **12.3** | |

## C   ANALYSIS OF SDE-BASED SAMPLING ALGORITHMS

In this section, we analyze the behavior of the SDE-based sampling methods Algs. 3 and 4.

### C.1   EQUIVALENCE OF THE TWO SAMPLING ALGORITHMS FOR INFINITE STEPS

It is worth noting that for $\Delta t \to 0$, the Ornstein-Uhlenbeck discretization and the Euler-Maruyama discretization of Eq. (7) converge to the same update step:

$$\mathbf{z}_{t+\Delta t} \sim m + (\mathbf{z}_t - m)e^{-\kappa \Delta t} + \sqrt{\frac{\gamma \beta'}{2\kappa}(1 - e^{-2\kappa \Delta t})} \cdot \mathcal{N}(0,1) \tag{14}$$

$$\to m + (\mathbf{z}_t - m)(1 - \kappa \Delta t) + \sqrt{\frac{\gamma \beta'}{2\kappa}(1 - (1 - 2\kappa \Delta t))} \cdot \mathcal{N}(0,1) \tag{15}$$

$$= \mathbf{z}_t + \kappa(m - \mathbf{z}_t)\Delta t + \sqrt{\gamma \beta' \Delta t} \cdot \mathcal{N}(0,1) \tag{16}$$

$$= \mathbf{z}_t + \kappa(\boldsymbol{\mu}_0 + (\beta + \beta'/\kappa)\hat{\mathbf{x}} - \mathbf{z}_t)\Delta t + \sqrt{\gamma \beta' \Delta t} \cdot \mathcal{N}(0,1) \tag{17}$$

$$= \mathbf{z}_t + \beta'\hat{\mathbf{x}}\Delta t + \frac{\gamma - 1}{2}\beta'\frac{\boldsymbol{\mu}_0 + \beta\hat{\mathbf{x}} - \mathbf{z}_t}{\beta + \beta_0}\Delta t + \sqrt{\gamma \beta' \Delta t} \cdot \mathcal{N}(0,1) \tag{18}$$

$$= \mathbf{z}_t + \beta' f_\theta(\mathbf{z}_t, t)\Delta t + \frac{\gamma - 1}{2}\beta'\nabla_{\mathbf{z}_t} \log p_t(\mathbf{z}_t)\Delta t + \sqrt{\gamma \beta' \Delta t} \cdot \mathcal{N}(0,1) \tag{19}$$

### C.2   STABILITY OF EULER-MARUYAMA SAMPLING

Let us explicitly write out the update step of the Euler-Maruyama discretization of Eq. (7):

$$\mathbf{z}_{t+\Delta t} \sim \mathbf{z}_t + \beta'\hat{\mathbf{x}}\Delta t + \frac{\gamma - 1}{2}\beta'\frac{\boldsymbol{\mu}_0 + \beta\hat{\mathbf{x}} - \mathbf{z}_t}{\beta + \beta_0}\Delta t + \sqrt{\gamma \beta' \Delta t} \cdot \mathcal{N}(0,1) \tag{20}$$

$$= \left(1 - \frac{(\gamma - 1)\beta'}{2(\beta + \beta_0)}\Delta t\right)\mathbf{z}_t + \beta'\hat{\mathbf{x}}\Delta t + \frac{(\gamma - 1)\beta'(\boldsymbol{\mu}_0 + \beta\hat{\mathbf{x}})}{2(\beta + \beta_0)}\Delta t + \sqrt{\gamma \beta' \Delta t} \cdot \mathcal{N}(0,1) \tag{21}$$

As a rule of thumb, the coefficient in front of $\mathbf{z}_t$ should not be negative, i.e., the previous step should not be over-corrected. This yields the condition

$$1 - \frac{(\gamma - 1)\beta'}{2(\beta + \beta_0)}\Delta t \geq 0 \tag{22}$$

$$\iff \Delta t \cdot (\gamma - 1) \leq \frac{2(\beta + \beta_0)}{\beta'} \tag{23}$$

For our precision schedule on moses ($\beta_{\texttt{start}} = 3.0, \beta_{\texttt{end}} = 12.0, \beta_0 = 1.0$), we find that

$$\min_{t \in [0,1]} \frac{2(\beta(t) + \beta_0)}{\beta'(t)} \approx 0.48 \tag{24}$$

The resulting maximum stable noise level $\gamma$ for different numbers of sampling steps in Tab. 4 predicts the observed behavior in Fig. 3 surprisingly well.

### C.3   BEHAVIOR OF ORNSTEIN-UHLENBECK SAMPLING WITH INFINITE NOISE

Taking the limit $\gamma \to \infty$ in Alg. 4 yields an interesting sampling algorithm (see Alg. 5). In this limit, the update step becomes independent of the previous step $\mathbf{z}_t$, replacing all previous information with the current prediction $\hat{\mathbf{x}}$. Empirically, we find that fixing the prior value after the initial sampling step, as shown in Alg. 6, works better in practice (see Tab. 5). This algorithm matches the Flowback algorithm from Song et al. (2025). We find that with a budget of 50 sampling steps, this algorithm performs surprisingly well on molecule generation. However, a higher compute budget drastically

Table 4: Maximum stable $\gamma$ for different numbers of sampling steps with the Euler-Maruyama discretization, following Eq. (24).

| Number of Timesteps | $\Delta t$ | Maximum Stable $\gamma$ |
| --- | --- | --- |
| 25 | 0.040000 | 12.938480 |
| 50 | 0.020000 | 24.876960 |
| 100 | 0.010000 | 48.753920 |
| 200 | 0.005000 | 96.507840 |
| 500 | 0.002000 | 239.769601 |

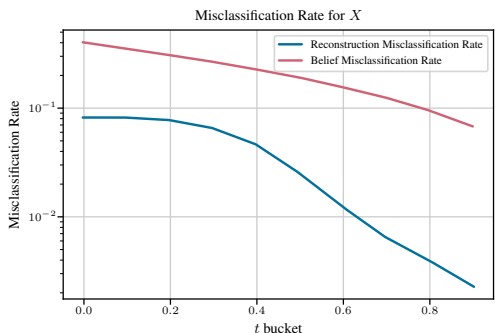

Figure 5: Empirical misclassification rate of a trained reconstructor on the moses dataset under the encoding distribution. Compared to simply sampling from the belief, returning a reconstruction is far more likely to yield the correct train sample. Therefore, returning a quantization of the reconstruction instead of sampling from the belief is significantly more efficient for molecule generation. However, deriving the ELBO under quantization is intractable to optimize. Therefore, we have the sampling-formulation to derive a tractable ELBO and the quantized-formulation to optimize efficiency after training.

reduces performance. We hypothesize that this is because an excessive amount of stochasticity is introduced. Song et al. (2025) address this by adaptively alternating between vanilla BFN steps and Flowback steps, effectively mixing Alg. 1 with Alg. 5.

---

**Algorithm 5** Sampling with $\gamma \to \infty$

**Require:** reconstructor $f_\theta$, discretization $\Delta t$, precision schedule $\beta : [0, 1] \to \mathbb{R}^+$
$\mathbf{z}_0 \sim \mathcal{N}(\boldsymbol{\mu}_0, \beta_0 I)$
$\mathbf{z} \leftarrow \mathbf{z}_0$
**for** $t = 0 \dots 1$ in steps of $\Delta t$ **do**
$\quad \hat{\mathbf{x}} \leftarrow f_\theta(\mathbf{z}, t)$
$\quad \alpha \leftarrow \beta_0 + \beta(t + \Delta t/2)$
$\quad \mathbf{y} \sim \mathcal{N}(\mu = \hat{\mathbf{x}}, \Sigma^2 = 1/\alpha \cdot I)$
$\quad \triangleright$ Go from prior to $t$ in single step
$\quad \mathbf{z} \leftarrow \boldsymbol{\mu}_0 + \alpha \cdot \mathbf{y}$
**end for**
**return** $\mathrm{Quantize}(f_\theta(\mathbf{z}, 1))$

---

**Algorithm 6** Fixed-prior sampling with $\gamma \to \infty$

**Require:** reconstructor $f_\theta$, discretization $\Delta t$, precision schedule $\beta : [0, 1] \to \mathbb{R}^+$
$\mathbf{z}_0 \sim \mathcal{N}(\boldsymbol{\mu}_0, \beta_0 I)$
$\mathbf{z} \leftarrow \mathbf{z}_0$
**for** $t = 0 \dots 1$ in steps of $\Delta t$ **do**
$\quad \hat{\mathbf{x}} \leftarrow f_\theta(\mathbf{z}, t)$
$\quad \alpha \leftarrow \beta(t + \Delta t/2)$
$\quad \mathbf{y} \sim \mathcal{N}(\mu = \hat{\mathbf{x}}, \Sigma^2 = 1/\alpha \cdot I)$
$\quad \triangleright$ Go from prior to $t$ in single step
$\quad \mathbf{z} \leftarrow \mathbf{z}_0 + \alpha \cdot \mathbf{y}$
**end for**
**return** $\mathrm{Quantize}(f_\theta(\mathbf{z}, 1))$

---

Table 5: Results on the GuacaMol and Moses benchmarks for molecular generation with the Euler-(EM) and Ornstein-Uhlenbeck (OU) discretization, and with Alg. 5 ($\gamma \to \infty$) and Alg. 6 ($\gamma \to \infty$, FP), as well as results for a linear scheduler (lin) with the same final precision as the exponential scheduler. Additionally, we include results obtained with the FlowBack (FB) sampler Song et al. (2025) using a smaller value of $\beta_0$, as well as the OU sampler with the same checkpoint. The EM sampler becomes unstable at $\beta_0 = 0.05$.

| Model | Steps | GuacaMol | | | | | Moses | | | | | | |
|---|---|---|---|---|---|---|---|---|---|---|---|---|---|
| | | Val. ↑ | V.U. ↑ | V.U.N. ↑ | KL ↑ | FCD ↑ | Val. ↑ | Uniq. ↑ | Nov. ↑ | Filters ↑ | FCD ↓ | SNN ↑ | Scaf ↑ |
| Train Set | | 100.0 | 100.0 | 0.0 | 99.9 | 92.8 | 100.0 | 100.0 | 0.0 | 100.0 | 0.01 | 0.64 | 99.1 |
| GraphBSI (EM) | 10 | 86.6 | 86.6 | 86.5 | 85.5 | 27.6 | 90.9 | 100.0 | 99.2 | 85.4 | 3.74 | 0.43 | 13.7 |
| GraphBSI (OU) | 10 | 91.9 | 91.9 | 91.8 | 84.5 | 24.2 | 94.4 | 100.0 | 98.9 | 89.0 | 3.88 | 0.45 | 14.5 |
| GraphBSI (EM) | 20 | 97.5 | 97.5 | 97.3 | 87.5 | 40.7 | 97.5 | 100.0 | 97.9 | 93.6 | 1.83 | 0.47 | 15.7 |
| GraphBSI (OU) | 20 | 97.1 | 97.1 | 96.8 | 89.3 | 49.7 | 98.2 | 100.0 | 97.8 | 94.5 | 1.92 | 0.48 | 14.4 |
| DeFoG | 50 | 91.7 | 91.7 | 91.2 | 92.3 | 57.9 | 83.9 | 99.9 | 96.9 | 96.5 | 1.87 | 0.50 | 23.5 |
| GraphBSI (EM) | 50 | 97.5 | 97.5 | 97.2 | 90.7 | 65.6 | 99.3 | 100.0 | 96.5 | 96.9 | 1.06 | 0.50 | 15.2 |
| GraphBSI (OU) | 50 | 99.2 | 99.2 | 98.7 | 93.7 | 71.3 | 99.7 | 100.0 | 94.6 | 98.2 | 1.19 | 0.52 | 15.1 |
| GraphBSI ($\gamma \to \infty$) | 50 | 99.6 | 99.6 | 98.3 | 95.1 | 61.4 | 99.9 | 99.9 | 89.9 | 99.2 | 1.58 | 0.56 | 11.7 |
| GraphBSI ($\gamma \to \infty$,FP) | 50 | 99.6 | 99.6 | 98.3 | 97.4 | 75.1 | 99.9 | 99.9 | 89.7 | 99.1 | 1.06 | 0.56 | 13.1 |
| GraphBSI (FB) | 50 | - | - | - | - | - | 99.6 | 100.0 | 95.9 | 97.5 | 1.15 | 0.51 | 15.0 |
| DiGress (CADD) | 500 | - | - | - | - | - | 92.2 | 82.3 | 74.2 | 76.2 | 37.19 | 0.24 | 0.0 |
| DiGress | 500 | 85.2 | 85.2 | 85.1 | 92.9 | 68.0 | 85.7 | 100.0 | 95.0 | 97.1 | 1.19 | 0.52 | 14.8 |
| DisCo | 500 | 86.6 | 86.6 | 86.5 | 92.6 | 59.7 | 88.3 | 100.0 | 97.7 | 95.6 | 1.44 | 0.50 | 15.1 |
| Cometh | 500 | 98.9 | 98.9 | 97.6 | 96.7 | 72.7 | 90.5 | 99.9 | 92.6 | 99.1 | 1.27 | 0.54 | 16.0 |
| DeFoG | 500 | 99.0 | 99.0 | 97.9 | 97.7 | 73.8 | 92.8 | 99.9 | 92.1 | 98.9 | 1.95 | 0.55 | 14.4 |
| GraphBFN | 500 | - | - | - | - | - | 98.5 | 99.8 | 89.0 | 98.3 | 1.07 | 0.59 | 10.0 |
| GraphBSI (EM) | 500 | 98.8 | 98.8 | 98.3 | 94.6 | 82.6 | 99.8 | 100.0 | 92.5 | 99.1 | 0.72 | 0.54 | 14.3 |
| GraphBSI (OU) | 500 | 99.6 | 99.6 | 98.2 | 98.4 | 80.3 | 99.9 | 100.0 | 90.7 | 99.2 | 0.90 | 0.55 | 12.7 |
| GraphBSI (EM,lin) | 500 | - | - | - | - | - | 99.8 | 100.0 | 91.9 | 99.1 | 0.85 | 0.54 | 12.3 |
| GraphBSI (OU,lin) | 500 | - | - | - | - | - | 99.9 | 99.9 | 90.6 | 99.2 | 0.98 | 0.55 | 14.2 |
| GraphBSI (FB,$\beta_0 = 1$) | 500 | - | - | - | - | - | 100.0 | 99.6 | 80.7 | 99.6 | 2.84 | 0.59 | 8.7 |
| GraphBSI (FB,$\beta_0 = 0.05$) | 500 | - | - | - | - | - | 100.0 | 99.9 | 85.9 | 99.4 | 1.32 | 0.57 | 11.7 |
| GraphBSI (OU,$\beta_0 = 0.05$) | 500 | - | - | - | - | - | 99.9 | 100.0 | 90.4 | 99.4 | 1.00 | 0.55 | 12.4 |

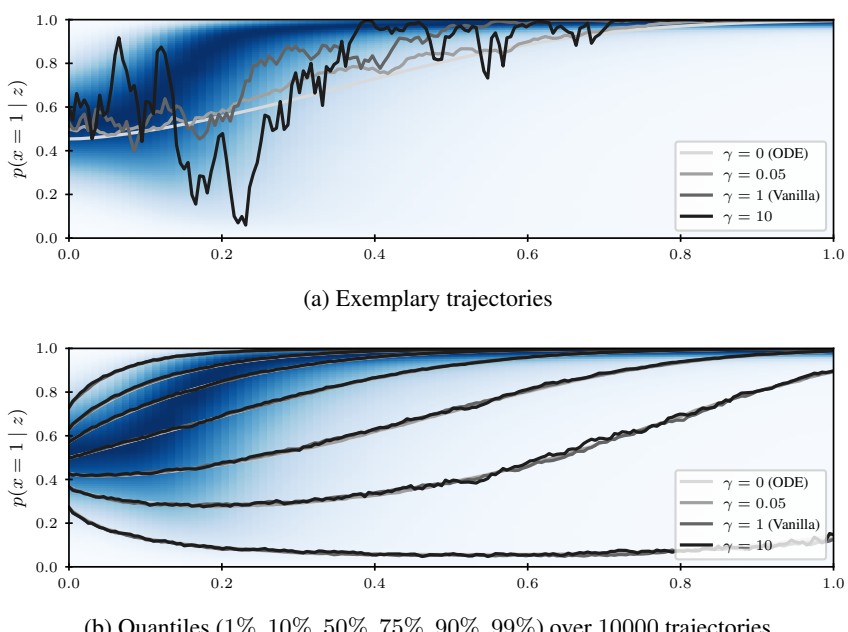

(a) Exemplary trajectories

(b) Quantiles ($1\%, 10\%, 50\%, 75\%, 90\%, 99\%$) over 10000 trajectories

Figure 6: Illustration of the trajectories of the categorical sampler with two categories with a fixed reconstruction $f(\mathbf{z}, t) = \hat{e}_1$ for different noise levels $\gamma$. While higher values of $\gamma$ result in more volatile trajectories (see Fig. 6a), the marginal distribution is preserved if the score function is known exactly (see Fig. 6b). Since we approximate the score function in practice, the noise level is a crucial hyperparameter to fin-tune during inference.

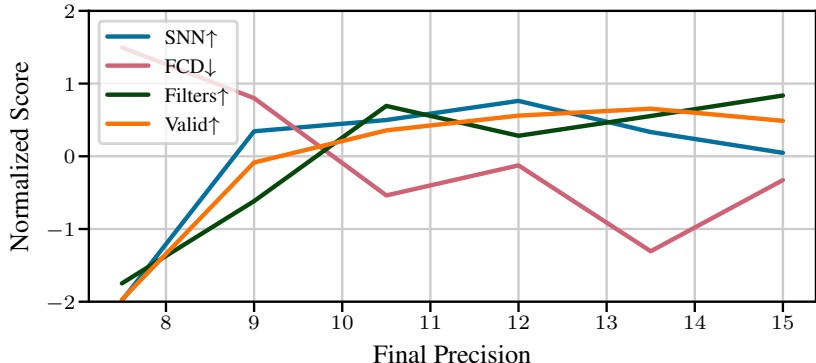

Figure 7: Key metrics on the Moses benchmark with a linear scheduler, ending at different final precisions. The model was trained with a final precision of 15, and to generate this plot, sampling was stopped early instead of training a new model for each precision value. While too small final precision values yield noisy samples, too large final precision values waste sampling steps.

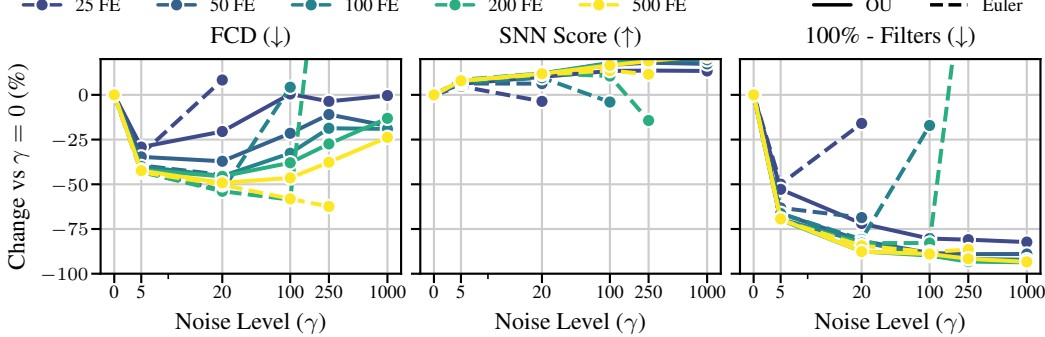

Figure 8: Change in metrics relative to $\gamma = 0$ vs. noise level $\gamma$ for different numbers of function evaluations (FE) and discretization schemes. Our custom Ornstein-Uhlenbeck discretization scheme is denoted as OU, while the standard Euler-Maruyama scheme is written as Euler. Some values for the Euler scheme are missing since the sampler becomes unstable if $\gamma \cdot \Delta t$ becomes too large (see Sec. C.2).

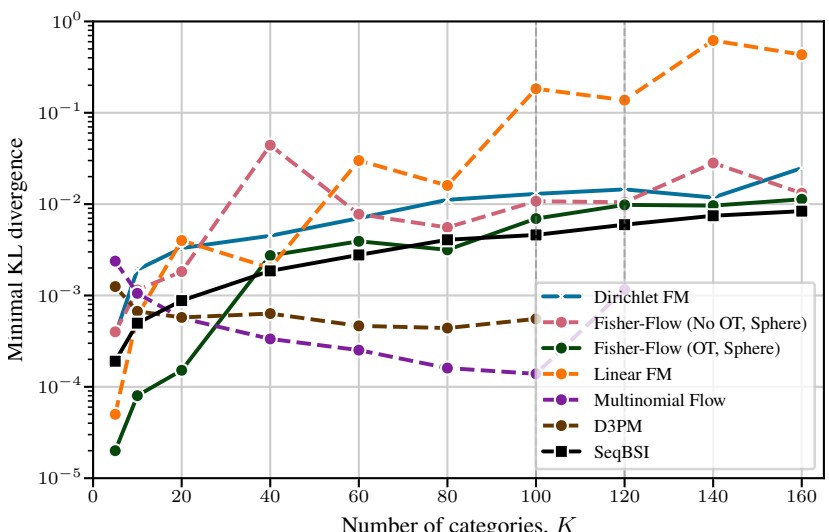

Figure 9: KL divergence on the toy sequences benchmark by (Davis et al., 2024), reporting the lowest KL divergence for each vocabulary size over five random seeds. The model is trained on $100,000$ samples with a sequence of length four and varying vocabulary size. Find the details of the dataset generation in the original paper.

# D PROOFS

**Theorem 1.** *Given a prior belief $p(\mathbf{x} \mid \mathbf{z}) = \mathrm{Cat}(\mathbf{x} \mid \mathrm{softmax}(\mathbf{z}))$, after observing $\mathbf{y} \sim \mathcal{N}(\mathbf{y} \mid \mu = \mathbf{x}, \Sigma^2 = 1/\alpha I)$ at precision $\alpha$, the posterior belief is $p(\mathbf{x} \mid \mathbf{z}, \mathbf{y}, \alpha) = \mathrm{Cat}(\mathbf{x} \mid \mathrm{softmax}(\mathbf{z}_{\mathrm{post}}))$ with*

$$\mathbf{z}_{\mathrm{post}} = \mathbf{z} + \alpha \cdot \mathbf{y} \tag{25}$$

*Proof.* We need to compute the Bayesian update of the belief parameters. Each dimension can be considered independently since the noise is isotropic. Let us start with a single-variable prior belief $\mathrm{Cat}(\mathrm{softmax}(\mathbf{z}))$ with $\mathbf{z} \in \mathbb{R}^c$, and a noisy observation $\mathbf{y} \mid \mathbf{x}, \alpha \sim \mathcal{N}(\mu = \mathbf{x}, \Sigma^2 = 1/\alpha \cdot I)$ of the true sample $\mathbf{x} \in \Delta^{c-1}$ at precision $\alpha$. Let us now consider any class $l \in 1, \ldots, c$. We write $\hat{e}_l$ for the one-hot encoding of class $l$. Since we are only interested in the ratio of the posterior probabilities, we can ignore any factors that do not depend on $l$ and normalize at the end. We have:

$$p(\mathbf{x} = \hat{e}_l \mid \mathbf{z}) = \mathrm{softmax}(\mathbf{z})_l \propto \exp(\mathbf{z}_l) \tag{26}$$

$$p(\mathbf{y} \mid \mathbf{x} = \hat{e}_l, \alpha) = \mathcal{N}(\mathbf{y} \mid \mu = \hat{e}_l, \Sigma^2 = 1/\alpha \cdot I) \tag{27}$$

$$p(\mathbf{x} = \hat{e}_l \mid \mathbf{z}, \mathbf{y}, \alpha) = \propto p(\mathbf{y} \mid \mathbf{x} = \hat{e}_l, \alpha) \cdot p(\mathbf{x} = \hat{e}_l \mid \mathbf{z}) \tag{28}$$

$$= \mathcal{N}(\mathbf{y} \mid \mu = \hat{e}_l, \Sigma^2 = 1/\alpha \cdot I) \cdot \mathrm{softmax}(\mathbf{z})_l \tag{29}$$

$$\propto \exp\left(-\frac{\|\mathbf{y} - \hat{e}_l\|^2}{2 \cdot 1/\alpha}\right) \cdot \exp(\mathbf{z}_l) \tag{30}$$

$$= \exp\left(-\frac{\|\mathbf{y}\|^2 - 2 \cdot \langle \mathbf{y}, \hat{e}_l \rangle + \|\hat{e}_l\|^2}{2 \cdot 1/\alpha} + \mathbf{z}_l\right) \tag{31}$$

$$\propto \exp\left(\alpha \cdot \mathbf{y}_l + \mathbf{z}_l\right) \tag{32}$$

Let us now normalize the results to obtain the posterior probabilities:

$$p(\mathbf{x} = \hat{e}_l \mid \mathbf{z}, \mathbf{y}, \alpha) = \frac{\exp\left(\alpha \cdot \mathbf{y}_l + \mathbf{z}_l\right)}{\sum_{l'=1}^c \exp\left(\alpha \cdot \mathbf{y}_{l'} + \mathbf{z}_{l'}\right)} = \mathrm{softmax}(\mathbf{z} + \alpha \cdot \mathbf{y})_l \tag{33}$$

Putting everything together, we find that the posterior belief is $p(\mathbf{x} \mid \mathbf{z}, \mathbf{y}, \alpha) = \mathrm{Cat}(\mathbf{x} \mid \mathrm{softmax}(\mathbf{z}_{\mathrm{post}}))$ with

$$\mathbf{z}_{\mathrm{post}} = \mathbf{z} + \alpha \cdot \mathbf{y} \tag{34}$$

$\square$

**Theorem 2.** *For categorical BSI, the log-likelihood of $\mathbf{x}$ under Alg. 1 is lower-bounded by*

$$\log p(\mathbf{x}) \geq \mathop{\mathbb{E}}_{\mathbf{z}_k \sim q(\mathbf{z}|\mathbf{x}, t_k)}[\log p(\mathbf{x} \mid \mathbf{z}_k)] - \frac{k}{2} \mathop{\mathbb{E}}_{\substack{i \sim \mathcal{U}(0, k-1) \\ \mathbf{z}_i \sim q(\mathbf{z}|\mathbf{x}, t_i)}}[(\beta(t_{i+1}) - \beta(t_i))\|f_\theta(\mathbf{z}_i, t_i) - \mathbf{x}\|_2^2], \tag{35}$$

*where $q(\mathbf{z} \mid \mathbf{x}, t) = \mathcal{N}(\mathbf{z} \mid \boldsymbol{\mu}_0 + \beta(t)\mathbf{x}, \beta_0 + \beta(t)I)$.*

*Proof.* For any distribution $p(\mathbf{x})$ and any latent variable $\mathbf{z}$, i.e. any choice of prior $p(\mathbf{z})$, encoding distribution $p(\mathbf{z} \mid \mathbf{x})$, and likelihood $p(\mathbf{x} \mid \mathbf{z})$, we have the variational lower bound

$$\log p(\mathbf{x}) \geq \mathop{\mathbb{E}}_{\mathbf{z} \sim p(\mathbf{z}|\mathbf{x})}[\log p(\mathbf{x} \mid \mathbf{z})] - \mathrm{KL}(p(\mathbf{z} \mid \mathbf{x}) \| p(\mathbf{z})) \tag{36}$$

on $\log p(\mathbf{x})$ Kingma & Welling (2013). We choose the beliefs $\mathbf{z}_0, \ldots, \mathbf{z}_k$ as latent variables at the discretized time steps $t_0, \ldots, t_k$. We choose the encoding distribution to be the distribution of the beliefs under Alg. 1 with the reconstruction network $f_\theta$ replaced by the true sample $\mathbf{x}$:

$$p(\mathbf{z}_0, \ldots, \mathbf{z}_k \mid \mathbf{x}) = \mathcal{N}(\mathbf{z}_0 \mid \boldsymbol{\mu}_0, \beta_0 I) \prod_{i=0}^{k-1} p(\mathbf{z}_{i+1} \mid \mathbf{z}_i, \mathbf{x}, t_i) \tag{37}$$

The transition distribution $p(\mathbf{z}_{i+1} \mid \mathbf{z}_i, \mathbf{x}, t_i)$ can be computed from Theorem 1:

$$\mathbf{z}_{i+1} = \mathbf{z}_i + \alpha_i \cdot \mathbf{y}_i \sim \mathbf{z}_i + \alpha_i \cdot \mathcal{N}(\mathbf{y} \mid \mu = \mathbf{x}, 1/\alpha_i I) = \mathcal{N}(\mathbf{z}_{i+1} \mid \mathbf{z}_i + \alpha_i \cdot \mathbf{x}, \alpha_i I) \tag{38}$$

The distribution of $p(\mathbf{z})$ following Alg. 1 factorizes similarly:

$$p(\mathbf{z}_0, \ldots, \mathbf{z}_k) = \mathcal{N}(\mathbf{z}_0 \mid \boldsymbol{\mu}_0, \beta_0 I) \prod_{i=0}^{k-1} p(\mathbf{z}_{i+1} \mid \mathbf{z}_i, t_i, \theta) \tag{39}$$

with the transition distribution

$$p(\mathbf{z}_{i+1} \mid \mathbf{z}_i, t_i, \theta) = \mathcal{N}(\mathbf{z}_{i+1} \mid \mathbf{z}_i + \alpha_i \cdot f_\theta(\mathbf{z}_i, t_i), \alpha_i I) \tag{40}$$

Let us now compute the KL divergence:

$$\mathrm{KL}(p(\mathbf{z}_0, \ldots, \mathbf{z}_k \mid \mathbf{x}) \| p(\mathbf{z}_0, \ldots, \mathbf{z}_k)) \tag{41}$$

$$= \mathop{\mathbb{E}}_{\substack{\mathbf{z}_0,\ldots,\mathbf{z}_k \sim \\ p(\mathbf{z}_0,\ldots,\mathbf{z}_k|\mathbf{x})}} \left[ \log \frac{p(\mathbf{z}_0, \ldots, \mathbf{z}_k \mid \mathbf{x})}{p(\mathbf{z}_0, \ldots, \mathbf{z}_k)} \right] \tag{42}$$

$$= \mathop{\mathbb{E}}_{\substack{\mathbf{z}_0,\ldots,\mathbf{z}_k \sim \\ p(\mathbf{z}_0,\ldots,\mathbf{z}_k|\mathbf{x})}} \left[ \log \frac{\mathcal{N}(\mathbf{z}_0 \mid \boldsymbol{\mu}_0, \beta_0 I) \prod_{i=0}^{k-1} p(\mathbf{z}_{i+1} \mid \mathbf{z}_i, \mathbf{x}, t_i)}{\mathcal{N}(\mathbf{z}_0 \mid \boldsymbol{\mu}_0, \beta_0 I) \prod_{i=0}^{k-1} p(\mathbf{z}_{i+1} \mid \mathbf{z}_i, t_i, \theta)} \right] \tag{43}$$

$$= \mathop{\mathbb{E}}_{\substack{\mathbf{z}_0,\ldots,\mathbf{z}_k \sim \\ p(\mathbf{z}_0,\ldots,\mathbf{z}_k|\mathbf{x})}} \left[ \sum_{i=0}^{k-1} \log \frac{p(\mathbf{z}_{i+1} \mid \mathbf{z}_i, \mathbf{x}, t_i)}{p(\mathbf{z}_{i+1} \mid \mathbf{z}_i, t_i, \theta)} \right] \tag{44}$$

$$= \sum_{i=0}^{k-1} \mathop{\mathbb{E}}_{\mathbf{z}_i \sim p(\mathbf{z}_i|\mathbf{x})} [\mathrm{KL}(p(\mathbf{z}_{i+1} \mid \mathbf{z}_i, \mathbf{x}, t_i) \| p(\mathbf{z}_{i+1} \mid \mathbf{z}_i, t_i, \theta))] \tag{45}$$

$$= \sum_{i=0}^{k-1} \mathop{\mathbb{E}}_{\mathbf{z}_i \sim p(\mathbf{z}_i|\mathbf{x})} [\mathrm{KL}(\mathcal{N}(\mathbf{z}_{i+1} \mid \mathbf{z}_i + \alpha_i \cdot \mathbf{x}, \alpha_i I) \| \mathcal{N}(\mathbf{z}_{i+1} \mid \mathbf{z}_i + \alpha_i \cdot f_\theta(\mathbf{z}_i, t_i), \alpha_i I))] \tag{46}$$

$$= \sum_{i=0}^{k-1} \mathop{\mathbb{E}}_{\mathbf{z}_i \sim p(\mathbf{z}_i|\mathbf{x})} \left[ \frac{1}{2\alpha_i} ||\mathbf{z}_i + \alpha_i \cdot \mathbf{x} - (\mathbf{z}_i + \alpha_i \cdot f_\theta(\mathbf{z}_i, t_i))||_2^2 \right] \tag{47}$$

$$= \sum_{i=0}^{k-1} \mathop{\mathbb{E}}_{\mathbf{z}_i \sim p(\mathbf{z}_i|\mathbf{x})} \left[ \frac{\alpha_i}{2} ||\mathbf{x} - f_\theta(\mathbf{z}_i, t_i)||_2^2 \right] \tag{48}$$

$$= \sum_{i=0}^{k-1} \mathop{\mathbb{E}}_{\mathbf{z}_i \sim p(\mathbf{z}_i|\mathbf{x})} [(\beta(t_{i+1}) - \beta(t_i))/2 ||\mathbf{x} - f_\theta(\mathbf{z}_i, t_i)||_2^2] \tag{49}$$

$$= \mathop{\mathbb{E}}_{\substack{i \sim \mathcal{U}(0,k-1) \\ \mathbf{z}_i \sim p(\mathbf{z}_i|\mathbf{x})}} \left[ \frac{k}{2} (\beta(t_{i+1}) - \beta(t_i)) ||\mathbf{x} - f_\theta(\mathbf{z}_i, t_i)||_2^2 \right] \tag{50}$$

Since $p(\mathbf{x} \mid \mathbf{z}_0, \ldots, \mathbf{z}_k) = p(\mathbf{x} \mid \mathbf{z}_k) = \mathrm{Cat}(\mathbf{x} \mid \mathrm{softmax}(\mathbf{z}_k))$, we can plug in Eq. (3) to obtain the final result:

$$\log p(\mathbf{x}) \geq \mathop{\mathbb{E}}_{\mathbf{z}_k \sim q(\mathbf{z}|\mathbf{x}, t_k)} [\log p(\mathbf{x} \mid \mathbf{z}_k)] - \frac{k}{2} \mathop{\mathbb{E}}_{\substack{i \sim \mathcal{U}(0,k-1) \\ \mathbf{z}_i \sim q(\mathbf{z}|\mathbf{x}, t_i)}} [(\beta(t_{i+1}) - \beta(t_i)) ||f_\theta(\mathbf{z}_i, t_i) - \mathbf{x}||_2^2], \tag{51}$$

where $q(\mathbf{z} \mid \mathbf{x}, t) = \mathcal{N}(\mathbf{z} \mid \boldsymbol{\mu}_0 + \beta(t)\mathbf{x}, \beta_0 + \beta(t)I)$. $\qquad\square$

**Theorem 3.** *As $\Delta t \to 0$, the update equation in Theorem 1 converges to the following SDE:*

$$d\mathbf{z}_t = \beta'(t) f_\theta(\mathbf{z}_t, t) dt + \sqrt{\beta'(t)} dW_t \tag{52}$$

*where $dW_t$ is a Wiener process and $\mathbf{z}_0 \sim \mathcal{N}(\boldsymbol{\mu}_0, \beta_0 \cdot I)$.*

*Proof.* Take the update equation Theorem 1 with an infinitesimal time step $\Delta t \to 0$, it holds that

$$\alpha = (\beta(t + \Delta t) - \beta(t)) \to \beta'(t)\Delta t \tag{53}$$

Therefore, we have:

$$\mathbf{z}_{t+\Delta t} = \mathbf{z}_t + \alpha \mathbf{y} \tag{54}$$

$$\sim \mathbf{z}_t + \alpha \mathcal{N}(\hat{\mathbf{x}}, \Sigma^2 = 1/\alpha I) \tag{55}$$

$$= \mathbf{z}_t + \mathcal{N}(\alpha \hat{\mathbf{x}}, \Sigma^2 = \alpha I) \tag{56}$$

$$\rightarrow \mathbf{z}_t + \beta'(t)\hat{\mathbf{x}}\Delta t + \sqrt{\beta'(t)}\sqrt{\Delta t} \cdot \mathcal{N}(0, \mathbf{I}) \tag{57}$$

We identify this as the Euler-Maruyama discretization of the SDE above. $\qquad \square$

**Theorem 4.** *The SDE in Theorem 3 is generalized by the following family of SDEs with equal marginal densities $p_t(\mathbf{z}_t)$:*

$$d\mathbf{z}_t = \beta'(t)f_\theta(\mathbf{z}_t, t)dt + \frac{\gamma - 1}{2}\beta'(t)\nabla_{\mathbf{z}_t}\log p_t(\mathbf{z}_t)dt + \sqrt{\gamma\beta'(t)}dW_t \tag{58}$$

*where $dW_t$ is a Wiener process and $\mathbf{z}_0 \sim p(\mathbf{z} \mid t = 0)$.*

*Proof.* We need to show that the evolution of the probability density $p_t(\mathbf{z}_t)$ of Eq. (6) matches that of Eq. (7). The evolution is characterized by the Fokker-Planck equation:

$$\frac{\partial p_t(\mathbf{z}_t)}{\partial t} = \sum_j -\nabla_{\mathbf{z}_j}\left(\beta'(t)f_\theta(\mathbf{z}_t, t) + \frac{\gamma - 1}{2}\beta'(t)\nabla_{\mathbf{z}_t}\log p_t(\mathbf{z}_t)\right)p_t(\mathbf{z}_t) + \frac{1}{2}\gamma\beta'(t)\nabla_{\mathbf{z}_j}^2 p_t(\mathbf{z}_t)$$

$$= \sum_j -\nabla_{\mathbf{z}_j}\left(\beta'(t)f_\theta(\mathbf{z}_t, t)p_t(\mathbf{z}_t)\right) - \frac{\gamma - 1}{2}\beta'(t)\nabla_{\mathbf{z}_j}\left(p_t(\mathbf{z}_t)\nabla_{\mathbf{z}_j}\log p_t(\mathbf{z}_t)\right) + \frac{1}{2}\gamma\beta'(t)\nabla_{\mathbf{z}_j}^2 p_t(\mathbf{z}_t)$$

$$= \sum_j -\nabla_{\mathbf{z}_j}\left(\beta'(t)f_\theta(\mathbf{z}_t, t)p_t(\mathbf{z}_t)\right) - \frac{\gamma - 1}{2}\beta'(t)\nabla_{\mathbf{z}_j}^2 p_t(\mathbf{z}_t) + \frac{1}{2}\gamma\beta'(t)\nabla_{\mathbf{z}_j}^2 p_t(\mathbf{z}_t)$$

$$= \sum_j -\nabla_{\mathbf{z}_j}\left(\beta'(t)f_\theta(\mathbf{z}_t, t)p_t(\mathbf{z}_t)\right) + \frac{1}{2}\beta'(t)\nabla_{\mathbf{z}_j}^2 p_t(\mathbf{z}_t)$$

Which equals the Fokker-Planck equation of the SDE in Eq. (6). $\qquad \square$

**Theorem 5.** *The BSI loss Eq. (5) also is a score matching loss with the score model $s_\theta(\mathbf{z}, t)$ parameterized as*

$$s_\theta(\mathbf{z}, t) \equiv \frac{\boldsymbol{\mu}_0 + \beta(t)f_\theta(\mathbf{z}, t) - \mathbf{z}}{\beta(t) + \beta_0} \stackrel{!}{\approx} \nabla_{\mathbf{z}}\log p_t(\mathbf{z}) \tag{59}$$

*Proof.* Score matching Song et al. (2021) is a generative model that learns to approximate the score function $\nabla_{\mathbf{z}}\log p_t(\mathbf{z})$ of a distribution $p_t(\mathbf{z})$ by minimizing the score matching loss:

$$\mathcal{L}_{\text{score}} \equiv \mathbb{E}_{t\sim\mathcal{U}(0,1)}[\lambda(t)\mathbb{E}_{p(\mathbf{x})}\mathbb{E}_{p_t(\mathbf{z}|\mathbf{x})}\left[\|s_\theta(\mathbf{z}, t) - \nabla_{\mathbf{z}}\log p_t(\mathbf{z} \mid \mathbf{x})\|_2^2\right]] \tag{60}$$

where $\lambda : [0, 1] \mapsto \mathbb{R}^+$ is a positive weighting function. The distribution $p_t(\mathbf{z} \mid \mathbf{x})$ is the distribution of the latent variable at time $t$ given the true sample $\mathbf{x}$. For categorical BSI, we have from Eq. (3):

$$p_t(\mathbf{z} \mid \mathbf{x}) = \mathcal{N}(\mathbf{z} \mid \boldsymbol{\mu}_0 + \beta(t)\mathbf{x}, (\beta_0 + \beta(t))I) \tag{61}$$

The score function of an isotropic Gaussian can be computed in closed form:

$$\nabla_{\mathbf{z}}\log\mathcal{N}(\mathbf{z} \mid \mu, \sigma^2 I) = \nabla_{\mathbf{z}}\left(-\frac{\|\mathbf{z} - \mu\|^2}{2\sigma^2}\right) = -\frac{\mathbf{z} - \mu}{\sigma^2} \tag{62}$$

$$\tag{63}$$

Plugging in the parameters of $p_t(\mathbf{z} \mid \mathbf{x})$, we find:

$$\nabla_{\mathbf{z}}\log p_t(\mathbf{z} \mid \mathbf{x}) = -\frac{\mathbf{z} - (\boldsymbol{\mu}_0 + \beta(t)\mathbf{x})}{\beta_0 + \beta(t)} = \frac{\boldsymbol{\mu}_0 + \beta(t)\mathbf{x} - \mathbf{z}}{\beta_0 + \beta(t)} \tag{64}$$

With the proposed score model parameterization $s_\theta(\mathbf{z}, t)$, we find:

$$\mathcal{L}_{\texttt{score}} = \mathbb{E}_{t \sim \mathcal{U}(0,1)}[\lambda(t)\mathbb{E}_{p(\mathbf{x})}\mathbb{E}_{p_t(\mathbf{z}|\mathbf{x})}\left[\|s_\theta(\mathbf{z}, t) - \nabla_{\mathbf{z}}\log p_t(\mathbf{z} \mid \mathbf{x})\|_2^2]\right] \tag{65}$$

$$= \mathbb{E}_{t \sim \mathcal{U}(0,1)}[\lambda(t)\mathbb{E}_{p(\mathbf{x})}\mathbb{E}_{p_t(\mathbf{z}|\mathbf{x})}\left[\left\|\frac{\boldsymbol{\mu}_0 + \beta(t)f_\theta(\mathbf{z}, t) - \mathbf{z}}{\beta(t) + \beta_0} - \frac{\boldsymbol{\mu}_0 + \beta(t)\mathbf{x} - \mathbf{z}}{\beta_0 + \beta(t)}\right\|_2^2\right]] \tag{66}$$

$$= \mathbb{E}_{t \sim \mathcal{U}(0,1)}[\lambda(t)\mathbb{E}_{p(\mathbf{x})}\mathbb{E}_{p_t(\mathbf{z}|\mathbf{x})}\left[\left\|\frac{\beta(t)(f_\theta(\mathbf{z}, t) - \mathbf{x})}{\beta(t) + \beta_0}\right\|_2^2\right]] \tag{67}$$

$$= \mathbb{E}_{t \sim \mathcal{U}(0,1)}[\lambda(t)\frac{\beta(t)^2}{(\beta(t) + \beta_0)^2}\mathbb{E}_{p(\mathbf{x})}\mathbb{E}_{p_t(\mathbf{z}|\mathbf{x})}\left[\|(f_\theta(\mathbf{z}, t) - \mathbf{x})\|_2^2\right]] \tag{68}$$

$$\tag{69}$$

Choosing the weighting

$$\lambda(t) = \beta'(t)\frac{(\beta(t) + \beta_0)^2}{2\beta(t)^2}, \tag{70}$$

we find that the score matching loss equals the BSI loss in Eq. (5). Therefore, the BSI loss in Eq. (5) is a score-matching loss with the weighting Sec. D and the score function $s_\theta(\mathbf{z}, t)$ parameterized as in Eq. (59).

$\square$

**Theorem 6.** *Fixing the prediction $\hat{\mathbf{x}} = f_\theta(\mathbf{z}_t, t)$ and the values $\beta = \beta(t + \Delta t/2)$, $\beta' = \beta'(t + \Delta t/2)$ in Eq. (7) in a time interval $[t, t + \Delta t]$ yields an Ornstein-Uhlenbeck process with the exact marginal*

$$\mathbf{z}_{t+\Delta t} \sim m + (\mathbf{z}_t - m)e^{-\kappa\Delta t} + \sqrt{\frac{\gamma\beta'}{2\kappa}(1 - e^{-2\kappa\Delta t})} \cdot \mathcal{N}(0, I), \tag{71}$$

*where $\kappa = \frac{(\gamma-1)\beta'}{2(\beta_0+\beta)}$, $m = \boldsymbol{\mu}_0 + (\beta + \beta'/\kappa)\hat{\mathbf{x}}$.*

*Proof.* The SDE in Eq. (7) with fixed parameters $\beta, \beta', \hat{x}$ is given as

$$d\mathbf{z}_t = \beta'\hat{\mathbf{x}}dt + \frac{\gamma - 1}{2}\beta'\nabla_{\mathbf{z}_t}\log p_t(\mathbf{z}_t)dt + \sqrt{\gamma\beta'}dW_t \tag{72}$$

where $dW_t$ is a Wiener process and $\mathbf{z}_t \sim p(\mathbf{z} \mid t)$. Let us insert Theorem 5 to obtain

$$d\mathbf{z}_t = \beta'\hat{\mathbf{x}}dt + \frac{\gamma - 1}{2}\beta'\frac{\boldsymbol{\mu}_0 + \beta f_\theta(\mathbf{z}_t, t) - \mathbf{z}_t}{\beta + \beta_0}dt + \sqrt{\gamma\beta'}dW_t \tag{73}$$

$$= \frac{(\gamma - 1)\beta'}{2(\beta_0 + \beta)}\left(\boldsymbol{\mu}_0 + \left(\beta + \frac{2(\beta_0 + \beta)}{\gamma - 1}\right)\hat{\mathbf{x}} - \mathbf{z}_t\right)dt + \sqrt{\gamma\beta'}dW_t \tag{74}$$

Setting $\kappa = \frac{(\gamma-1)\beta'}{2(\beta_0+\beta)}$ and $m = \boldsymbol{\mu}_0 + (\beta + \beta'/\kappa)\hat{\mathbf{x}}$, we find

$$d\mathbf{z}_t = \kappa(m - \mathbf{z}_t)dt + \sqrt{\gamma\beta'}dW_t \tag{75}$$

which is an Ornstein-Uhlenbeck process. The exact marginal distribution of an Ornstein-Uhlenbeck process is given as Uhlenbeck & Ornstein (1930):

$$\mathbf{z}_{t+\Delta t} \sim m + (\mathbf{z}_t - m)e^{-\kappa\Delta t} + \sqrt{\frac{\gamma\beta'}{2\kappa}(1 - e^{-2\kappa\Delta t})} \cdot \mathcal{N}(0, I) \tag{76}$$

$\square$

## E  ADDITIONAL RESULTS

Tab. 6 shows our method is competitive on the QM9 dataset with removed hydrogen atoms, achieving state-of-the-art results on validity and FCD. We explicitly model charges on the nodes, enabling high validity scores.

Table 6: Results on the QM9 dataset.

| Model | Steps | QM9 (without H) | | | QM9 (with H) | | |
|---|---|---|---|---|---|---|---|
| | | Val. ↑ | Uniq. ↑ | FCD ↓ | Val. ↑ | Uniq. ↑ | FCD ↓ |
| Train Set | | 99.3 | 100.0 | 0.05 | 99.3 | 100.0 | 0.05 |
| DiGress | 500 | 99.0 | 96.2 | - | $95.4 \pm 1.1$ | $\mathbf{97.6 \pm 0.4}$ | - |
| DiGress (CADD) | 500 | 96.3 | 83.4 | 5.25 | - | - | - |
| DisCo | 500 | $99.3 \pm 0.6$ | - | - | - | - | - |
| Fisher FM | 500 | 95.3 | - | - | - | - | - |
| Cometh | 500 | $\underline{99.6 \pm 0.1}$ | $\mathbf{96.8 \pm 0.2}$ | $0.25 \pm 0.01$ | - | - | - |
| DeFoG | 50 | $98.9 \pm 0.1$ | $96.2 \pm 0.2$ | $0.26 \pm 0.00$ | - | - | - |
| DeFoG | 500 | $99.3 \pm 0.0$ | $\underline{96.3 \pm 0.3}$ | $\underline{0.12 \pm 0.00}$ | $98.0 \pm 0.0$ | $96.7 \pm 0.0$ | $\mathbf{0.05 \pm 0.00}$ |
| Ours | 50 | $\mathbf{99.9}$ | 93.7 | 0.30 | - | - | - |
| Ours | 500 | $\mathbf{99.9}$ | 96.2 | $\mathbf{0.09}$ | 99.8 | 96.6 | $\underline{0.08}$ |

Table 7: Hyperparameters used for the results in Tabs. 1 and 2. The precision schedule is parameterized as $\beta(t) = \beta_{\text{start}} \cdot (\exp(t \cdot \log(\beta_{\text{end}}/\beta_{\text{start}})) - 1)$.

| Dataset | Belief Parameters | | | Sampler 10% steps | | Sampler 100% steps) | |
|---|---|---|---|---|---|---|---|
| | $\beta_{\text{start}}$ | $\beta_{\text{end}}$ | $\beta^{(0)}$ | $\gamma$ (OU) | $\gamma$ (Euler) | $\gamma$ (OU) | $\gamma$ (Euler) |
| GuacaMol | | 12.0 | | 20.0 | 10.0 | 200.0 | 200.0 |
| Moses | | | | 10.0 | 20.0 | 90.0 | 120.0 |
| Planar | 3.0 | | 1.0 | | | | 200.0 |
| SBM | | 20.0 | | - | | 200.0 | 200.0 |
| Tree | | | | | | | 100.0 |

Table 8: Datasets with training samples and maximum number of nodes. For Moses, we use the `test_scaffolds` split for benchmarking, which is the standard test split.

| Dataset | Train samples | Max. Nodes |
|---|---|---|
| GuacaMol (Brown et al., 2019) | 1.3M | 88 |
| Moses (Polykovskiy et al., 2020) | 1.6M | 30 |
| Planar (Martinkus et al., 2022) | 128 | 64 |
| SBM (Martinkus et al., 2022) | 128 | 187 |
| Tree (Bergmeister et al., 2024) | 128 | 64 |

Table 9: Molecular metrics

| Metric | Short | Description |
|---|---|---|
| Validity | Val. | The fraction of generated molecules that are chemically valid according to RDKit. |
| Uniqueness | Uniq. | The number of unique molecules generated (counting permutations as the same molecule) divided by the total number of generated molecules when generating 10,000 molecules. |
| Novelty | Nov. | The fraction of generated molecules that are not present in the training set. |
| Valid & Unique | V.U. | The fraction of generated molecules that are both valid and unique. |
| Valid, Unique & Novel | V.U.N. | The fraction of generated molecules that are valid, unique, and novel. |
| KL Divergence | KL. | The normalized KL-Divergence between the distributions of various physicochemical descriptors between the generated set and the training set. |
| Fréchet ChemNet Distance (Moses) | FCD | Distance between the distributions of learned features of the generated molecules and those of the validation set, as computed by a pretrained ChemNet model. |
| Fréchet ChemNet Distance (GuacaMol) | FCD | Same as for Moses, but normalized with the transform $x \to \exp(-0.2x)$ |
| Similarity to Nearest Neighbor | SNN | The average Tanimoto similarity between each generated molecule and its nearest neighbor in the test set |
| Scaffold Similarity | Scaf. | Cosine similarity between the frequencies of scaffold substructures in the generated set and the test set |

Table 10: Synthetic graph metrics metrics

| Metric | Short | Description |
|---|---|---|
| Valid & Unique | V.U. | The fraction of generated graphs that are both valid and unique among 40 generated graphs. For the planar and tree datasets, we check if the generated graphs are planar/tree graphs. The SBM dataset does not have a straightforward validity criterion, therefore a test with Bayesian inference is used with a likelihood threshold. |
| Average Ratio | Ratio | For each of several metrics, *ratio* is defined as the Maximum Mean Discrepancy (MMD) between the generated and training set divided by the MMD between the training set and the test set. The average ratio is the ratio metric averaged over all metrics. The metrics are degree, clustering coefficient, orbit counts, spectral-, and wavelet metrics. |

