# OpenReview forum: "Discrete Bayesian Sample Inference for Graph Generation"
_ICLR.cc/2026/Conference — ICLR 2026 Poster_

### Official Review · Reviewer_WtYP · 2025-10-21

**Soundness:** 2
**Presentation:** 3
**Contribution:** 2
**Rating:** 4
**Confidence:** 4

**Summary:**

This paper introduces Discrete Bayesian Sampling Inference (BSI), a novel framework for graph generation. The core idea is to model the generative process not through discrete state transitions, but by evolving a continuous belief state, which is argued to better capture the dynamics of discrete variable evolution. The authors provide a theoretically grounded framework to derive the training and sampling methodologies. GraphBSI is evaluated on molecular graph generation benchmarks, where it demonstrates significant performance improvements over current state-of-the-art models.

**Strengths:**

- The proposed Discrete BSI presents a compelling and theoretically interesting mechanism for discrete data generation. The idea of operating on a continuous belief state to model the evolution of discrete variables addresses a key challenge in generative modeling.

- The application of BSI to graph generation yields significant performance improvements over state-of-the-art methods on established benchmarks like Moses and GuacaMol. This highlights the practical efficacy of the approach.

- The theoretical foundations are presented rigorously, with (mostly) well-stated theorems and proofs that are easy to follow.

**Weaknesses:**

- My primary concern is the precise positioning of the paper's contribution. It is somewhat ambiguous whether the main novelty lies in the general framework for discrete generation (placing it alongside with discrete diffusion, flow models, etc.) or specifically in its application to graph generation. Clarifying this would significantly strengthen the paper's impact by helping the reader understand its broader context and significance. Honestly I would be happy to improve the score if this question is properly addressed.

- Insufficient Ablation of Performance Gains: While the empirical gains are impressive, the source of these improvements is not fully elucidated. The provided ablation studies on noise levels and time distortion are helpful, but a deeper analysis is needed to isolate which specific components of the GraphBSI model are most critical to its success.

- The theoretical framework is solid but I would appreciate if the authors can provide some easy-to-follow explanations alongside. Some of the results are slightly against intuition (e.g. Theorem 2 where the lower bound is a squared error for categorical variables) and can be difficult to understand at the first glimpse. Also, I would appreciate if the authors can compare their methods with a few existing continuous state discrete diffusion/flow models, both theoretical and empirical. Such as [1, 2, 3].

[1] H. Stark. et al. "Dirichlet Flow Matching with Applications to DNA Sequence Design". ICML 2024

[2] O. Davis. et al. "Fisher flow matching for generative modeling over discrete data". NeurIPS 2024

[3] H. Zheng. et al. "Continuously augmented discrete diffusion model for categorical generative modeling". Arxiv.

**Questions:**

- Q1: Though the title states that the model is designated for graph generation, my main question is whether the model is specifically working for graph generation, or it can bring similar improvements to other discrete generation tasks. If this is designed for graph generation, I would like to see more on how can BSI improve the graph generation? If not and BSI can generally improve the discrete generation, it might be ideal to conduct generative tasks to modalities other than graphs, such as text modeling, protein co-design, etc[1][2].

- Q2: I look into the appendix and understand that the loss in eq. 5 is derived through the lower bound of log-likelihood. Honestly this is a bit surprised as from the assumption, the x is sampled from a softmax version of z, it seems cross-entropy loss to be more reasonable. Could the authors elaborate more on advantage of choosing the loss to be squared distance ? I am convinced with the derivation but It just seems a bit unnatural.

- Q3: In the abstract, the authors claim that the advantage of BSI comes from introducing a continuous belief z that can better capture the evolution of dynamics. Would the authors provide more evidence/justification/intuition on why this works?


[1] A. Campbell et al. A Continuous Time Framework for Discrete Denoising Models. NeurIPS 2023

[2] A. Campbell et al. Generative Flows on Discrete State-Spaces: Enabling Multimodal Flows with Applications to Protein Co-Design. ICML 2024

---

> ### Author Response · Authors · 2025-11-24
>
> We thank the reviewer for their valuable feedback. In the following, we address their comments.
>
> **Comment:** My primary concern is the precise positioning of the paper's contribution. It is somewhat ambiguous whether the main novelty lies in the general framework for discrete generation (placing it alongside with discrete diffusion, flow models, etc.) or specifically in its application to graph generation. Clarifying this would significantly strengthen the paper's impact by helping the reader understand its broader context and significance. Honestly I would be happy to improve the score if this question is properly addressed.
> **Response:** Our contributions are twofold. First, we extend the BSI framework to discrete data with a detailed theoretical comparison to BFNs. Second, we demonstrate this framework for graph generation. The **key** contribution lies in the general BSI formulation for discrete variables, which enables several sampling algorithms within a unified framework. While the theory is not restricted to graphs, we focus on (molecular) graph generation, as strong BFN and discrete diffusion baselines exist, allowing us to isolate the effect of our modifications. Furthermore, our modifications do not alter the ELBO, so likelihood-based evaluations remain unchanged across samplers, whereas the proposed sampling schemes primarily affect generative metrics. To clarify this point, we have added an experiment that generates DNA sequences with categorical BSI. We also updated the contributions section, emphasizing that categorical BSI can be applied to various types of discrete data, and we demonstrate this on graphs and sequences.
>
> **Comment:** Insufficient Ablation of Performance Gains: While the empirical gains are impressive, the source of these improvements is not fully elucidated. The provided ablation studies on noise levels and time distortion are helpful, but a deeper analysis is needed to isolate which specific components of the GraphBSI model are most critical to its success.
> **Response:** The key component we identified as a source of improvement is adapting the noise level. While changing the time distortion can help reduce the FCD, the changes are minor compared to the changes obtained by tuning the noise level. Find an additional ablation study in Figure 8. We further show that the effect of using an exponential instead of a linear precision schedule is marginal in Table 5. What matters about the precision schedule is that the final precision at $t=1$ is just high enough so that the reconstructor can predict clean train samples perfectly at $t=1$ (see Figure 5).
>
>
>
> **Comment:** The theoretical framework is solid but I would appreciate if the authors can provide some easy-to-follow explanations alongside. Some of the results are slightly against intuition (e.g. Theorem 2 where the lower bound is a squared error for categorical variables) and can be difficult to understand at the first glimpse. Also, I would appreciate if the authors can compare their methods with a few existing continuous state discrete diffusion/flow models, both theoretical and empirical. Such as Dirichlet Flow Matching (DFM), Fisher Flow Matching (FFM), and Continuously Augmented Discrete Diffusion (CADD).
>
> In terms of methodology, all three methods have various differences from ours. DFM and FFM both define a flow matching model with paths evolving on the simplex. Their corresponding models are trained by learning the corresponding vector fields via minimizing the conditional flow matching loss. Further, the generated samples are obtained by integrating their neural ODE. Our model, on the other hand, defines an SDE in the logit space and is trained by optimizing the ELBO. CADD combines discrete and continuous states. They have a discrete diffusion model augmented with a continuous diffusion, which generates the embedding vectors of the corresponding classes.
>
> Unfortunately, the code of CADD is not public yet, making a direct empirical comparison hard. However, we have contacted the authors, and they provided guidance on how to adjust existing discrete masking models to include their proposed continuous state. We have implemented a CADD version of DiGress, as their code base already includes an absorbing transition model. We have extended Table 5 in the appendix with these results. We have observed that compared to the original DiGress model, the validity improves while uniqueness, novelty, and other metrics worsen. We assume this is due to the absorbing transition.
>
> To compare to DFM, we have included an experiment on DNA sequence generation (see App. B).

---

> ### Author Response · Authors · 2025-11-24
>
> **Comment:** [...] [is] the model specifically working for graph generation, or it can bring similar improvements to other discrete generation tasks. If this is designed for graph generation, I would like to see more on how can BSI improve the graph generation? If not and BSI can generally improve the discrete generation, it might be ideal to conduct generative tasks to modalities other than graphs, such as text modeling, protein co-design, etc.
>
> **Response:** Our model is designed for graphs, but the derived theory can potentially be extended to other discrete generation tasks.
>
> We opted for graph generation for two main reasons. First, BFNs have already been successfully applied to graph generation [1], enabling the easier isolation of important components and direct comparison with various baselines. Second, graphs are a challenging discrete domain that requires modelling both nodes and edges. Further, many generative models have been tested on graphs before, e.g., [2,3,4], which all use similar architectures.
>
> Regarding how BSI benefits graph generation, the key contributions of our theoretical derivations are the generalizations of the sampling procedures, e.g., Eq. 7, Eq. 9, Eq. 10. In our empirical evaluations, these substantially improve results over our vanilla model (see Fig. 3). However, these improvements are of generative nature and do not affect likelihood-based metrics such as the perplexity or BPD, as the ELBO itself is unchanged.
>
> Finally, we also included new results applying our framework to DNA sequence generation (see App. B), as well as a theoretical comparison with Dirichlet Flow Matching in Appendix A.
>
> **Comment:** Why is the lower bound leading to a squared loss rather than a cross-entropy loss and what are the advantages?
>
> **Response:** The squared loss results from our likelihood optimization as our ELBO is expressed with the Gaussian latent variables $z_t$, rather than categorical variables. Therefore, the KL divergences are between the Gaussian beliefs over logits.
>
> While the cross-entropy loss may seem more reasonable at first, we would like to emphasize that, unlike discrete diffusion approaches, we do not operate on categorical distributions but on distributions over categorical distributions. Therefore, our ELBO quantifies the divergence between the distributions parametrizing the categorical distributions, i.e., the normal distributions.
>
> Intuitively, if we have an invertible change of variables, the KL divergence of two distributions is invariant under this change. Therefore, the KL divergence of two softmax-normalized random variables is equal to the KL divergence of the random variables themselves under certain conditions, e.g., normalized logits. Our setting is more general (i.e., logits are not normalized), but this explains how a squared loss results from the KL divergence when having distributions over categorical distributions.
>
> **Comment:**  In the abstract, the authors claim that the advantage of BSI comes from introducing a continuous belief $\mathbf{z}$ that can better capture the evolution of dynamics. Would the authors provide more evidence/justification/intuition on why this works?
> **Response:**
> The intuition behind it is that, unlike discrete state generative models, our neural network contains knowledge of the current state of each variable. For example, instead of making a binary decision whether an edge should be removed, i.e., flipping the edge to class 0, our model can slowly converge towards that state. In the discrete case, however, the edge would be removed within a single sampling step.
>
> Empirically, Table 1 shows GraphBSI outperforms DeFoG, a discrete state model with the same architecture and neural function evaluations.
>
> In addition, we conducted an ablation where the network is parametrized with a discrete graph while keeping the remaining parameters constant. More specifically, we use the argmax for each node and class instead of logits while keeping our training procedure, theory, and evaluation consistent. This experiment led to degenerate samples with around 1% uniqueness.
>
> We thank the reviewer again for their feedback and are happy to address any remaining concerns.
>
> [1] **Song, Y., Shi, J., Gong, J., Xu, M., Ermon, S., Zhou, H., & Ma, W. Y.** Smooth Interpolation for Improved Discrete Graph Generative Models. In Forty-second International Conference on Machine Learning.
>
> [2] **Vignac, C., Krawczuk, I., Siraudin, A., Wang, B., Cevher, V., & Frossard, P.** (2022). Digress: Discrete denoising diffusion for graph generation. arXiv preprint arXiv:2209.14734.
>
> [3] **Eijkelboom, F., Bartosh, G., Andersson Naesseth, C., Welling, M., & van de Meent, J. W.** (2024). Variational flow matching for graph generation. Advances in Neural Information Processing Systems, 37, 11735-11764.
>
> [4] **Qin, Y., Madeira, M., Thanou, D., & Frossard, P.** (2024). Defog: Discrete flow matching for graph generation. arXiv preprint arXiv:2410.04263.

---

> > ### Comment · Reviewer_WtYP · 2025-11-27
> >
> > Thank you for your efforts in the rebuttal. The response has sufficiently addressed my concern and I would like to increase the rating. Overall, I believe this paper presents a theoretically grounded framework to enhance discrete BSI with a continuous variable, which seems promising. It also underpins the foundation of discere diffusion/flow with a latent variable. I believe this paper can be further improved if the authors can provide more evidence on how this method benefits general discrete generation tasks beyond graph generation.

---

> > > ### Author Response · Authors · 2025-11-27
> > >
> > > We thank the reviewer for their positive response and for adjusting the score.
> > >
> > > We agree that additional experiments evaluating tasks beyond graph generation will enhance the paper. We are currently working on additional discrete-generation experiments to further compare with Fisher flow matching. If these results are ready during the discussion period, we will share them here. Otherwise, we will include them in the final version of the manuscript.

---

> ### Author Response · Authors · 2025-12-03
>
> We thank the reviewer for their additional feedback. We trained a Categorical BSI model on the synthetic sequence generation benchmark proposed in [1] to compare with Fisher Flow Matching [1] and Dirichlet Flow Matching [2] (see Fig. 9). Our results illustrate that Categorical BSI outperforms Dirichlet Flow Matching consistently and is competitive with Fisher Flow Matching.
>
> We again thank the reviewer for their valuable feedback.
>
> [1] *Davis, O., Kessler, S., Petrache, M., Ceylan, İ. İ., Bronstein, M., & Bose, A. J.* (2024). Fisher flow matching for generative modeling over discrete data. Advances in Neural Information Processing Systems, 37, 139054-139084.

---

### Official Review · Reviewer_skZ3 · 2025-10-27

**Soundness:** 2
**Presentation:** 3
**Contribution:** 2
**Rating:** 4
**Confidence:** 4

**Summary:**

The paper proposed a method based on BSI for graph generation, achieving outstanding performance.

**Strengths:**

The paper clearly presents the method being used. It is easy to follow.

It has a thorough structure, including theoretical guarantees and experimental analysis.

The experimental results are impressive, especially with only 50 steps.

**Weaknesses:**

Regarding the soundness of the paper, it is not clear how this work relates to graphs, and it seems mainly adapted for discrete data, and the transformers and graph features mainly follow the previous work. I would like to ask the authors to clarify whether they are the first to adapt BSI to discrete data modalities. If yes, is the main approach is adding softmax operations on the original feature z used in common BSI?

Beyond that, I kindly ask the authors to clarify better the relationship between BSI and BFN. The paper states that BFN is a more generalizable version of BSI, but a more thorough discussion would help the reader understand the connection better. In terms of application, how does this help in your implementation or analysis, and what can BFN not achieve?

Additionally, mathematically, what is the difference between this method and a continuous diffusion on the simplex space, or a continuous diffusion on the logits space with softmax applied? If different, what makes BFN more advantageous than these normal diffusion formulations?

Clarifying these points would help me better understand the contribution of the paper.

**Questions:**

1. Which preprocessing procedure (which version of the implementation) the authors apply for the MOSES dataset?
2. Do you have results with 5 or 10 steps, since the method already reaches very high performance with only 50 steps?
3. How do the authors evaluate the robustness of the framework, separately in terms of hyperparameters during training, and sampling?

---

> ### Author Response · Authors · 2025-11-24
>
> We thank the reviewer for their thorough feedback. In the following we address their questions.
>
> **Comment:** Regarding the soundness of the paper, it is not clear how this work relates to graphs, and it seems mainly adapted for discrete data, and the transformers and graph features mainly follow the previous work. I would like to ask the authors to clarify whether they are the first to adapt BSI to discrete data modalities. If yes, is the main approach adding softmax operations on the original feature z used in common BSI?
> **Response:**
> While our derived theory is not limited to graphs, we opted for graph generation for various reasons. First, the most related methodologies have been successfully applied to graphs, which includes GraphBFN, DeFog, and DiGress. This allows us to more easily isolate the performance gains achieved by our proposed framework. Second, while the discrete BSI can be applied to other discrete data such as language, the graph benchmarks provide an external evaluation for the generative performance. Our proposed framework has no substantial influence on likelihood-based evaluations, e.g., the perplexity, are not affected by our SDE based sampling algorithms.
>
> **Additional Results:** We also have included new results applying our framework to DNA sequence generation (see App. B)
> **Comparison to continuous BSI:** While our derived discrete BSI shares certain similarities to the continuous case, there are important differences which go beyond adding a softmax.
> Our updates on $\mathbf{x}$ are derived from a Bayesian update of a categorical distribution with a Gaussian likelihood, leading to Gaussian updates in the logit space $\mathbf{z}$. However, this update differs from the continuous case. I.e., our derived update: $p(z_t | z_{t-1}, x) = \mathcal{N}(z_{t-1} + \alpha_t x, \alpha_t I)$, compared to the continuous case: $p(z_t | z_{t-1}, x) = \mathcal{N}((\beta(t) z_{t-1} + \alpha_t x)/(\beta(t) + \alpha_t), (\alpha_t + \beta(t))^{-1} I)$. They are fundamentally different as our algorithm grows in magnitude across updates while the continuous case maintains is magnitude, resulting in a high entropy after applying a softmax.
>
> There has been a concurrent work [1], which derives the discrete Bayesian Flow Network updates using the notation of BSI, however, they neither generalize the update equations nor derive an ELBO. Their focus lays on hierarchy generation.
>
> We clarified that the update differs and added the missing related work in the updated manuscript.
>
> **Comment:** I kindly ask the authors to clarify better the relationship between BSI and BFN. The paper states that BFN is a more generalizable version of BSI, but a more thorough discussion would help the reader understand the connection better. In terms of application, how does this help in your implementation or analysis, and what can BFN not achieve?
> **Response:**
> BSI is a generalized version of BFN. More specifically, the dynamics of BFNs are recovered when choosing the sampler in equation 7 with $\gamma=1$ and $\beta_0=0$ to parametrize $z_0$, i.e., making the prior logits deterministic. Note that we require $\beta_0>0$ to avoid numerical issues when approximating the score function (see Eq. 8). Furthermore, BSI includes the sampler in Eq. 9, which are not within the previous framework.
>
> This generalized SDE allows BSI to vary stochasticity. This is visualized in the illustration in Figure 2, where $\gamma=1$ represents the dynamics of a standard BFN. Intuitively, increasing stochasticity allows the model to overwrite errors from previous predictions (see appendix B.3 for a discussion on the extreme case), and empirically, increasing stochasticity proves crucial for performance (see Fig. 3).
>
> We have extended App. A.1 and included this discussion.
>
>
> [1] **Kollovieh, M., Fleischmann, N., Guerranti, F., Charpentier, B., & Günnemann, S.** TreeGen: A Bayesian Generative Model for Hierarchies. *Advances in Neural Information Processing Systems, 38*

---

> ### Author Response · Authors · 2025-11-24
>
> **Comment:** Additionally, mathematically, what is the difference between this method and a continuous diffusion on the simplex space, or a continuous diffusion on the logits space with softmax applied? If different, what makes BFN more advantageous than these normal diffusion formulations?
> **Response:**
> Our SDE formulation obtained from the Bayesian update in Theorem 3 shares similarities with a diffusion model in the logit space.
> To compare these, we can derive a "noising" process, i.e., the posterior $p(\mathbf{z}_{t-1}\mid \mathbf{z}_t,\mathbf{x})$, which conceptually matches a forward diffusion process:
>
> $$p(z_t | z_{t+1}, x)
>     = \mathcal{N}\left(
>         \frac{(\beta_0 + \beta(t)) z_{t+1} + \alpha_t \mu_0 - \alpha_t \beta_0 x}
>              {\beta_0 + \beta(t) + \alpha_t},
>         \frac{\alpha_t (\beta_0 + \beta(t))}
>              {\beta_0 + \beta(t) + \alpha_t}\,I
>     \right),$$
>
> which is Gaussian, just like in standard diffusion models (see App A.1 in the updated manuscript for a derivation).
>
> The key difference to standard diffusion models is that the noising process explicitly depends on $\mathbf{x}$, making it non-Markovian in general. Interestingly, a Markovian transition is recovered in the special case $\beta_0=0$, which matches the original BFN parametrization.
>
> This, as mentioned in the previous response, would lead to a division by 0 in the approximation of the score function (Eq. 8), not allowing for the derivation of the family of SDEs with tunable stochasticity, which proved to be crucial for performance.
>
> We have included this discussion in App A.2.
>
> **Comment:** Which preprocessing procedure (which version of the implementation) the authors apply for the MOSES dataset?
> **Response:** Thank you for spotting the missing preprocessing description. We parse the MOSES smiles with RDKit and construct the graph features (X,E) out of the RDKit molecules. Here, the atom- and edge types are directly mapped to node- and edge categories, where the first edge type indicates the absence of an edge. We updated Section 4.1 with an explanation. We are happy to provide any further details on our preprocessing procedure.
>
> **Comment:** Do you have results with 5 or 10 steps, since the method already reaches very high performance with only 50 steps?
> **Response:** We have included results for 20 and 10 steps in Table 5. While 20 steps still work reasonably well, performance drops for 10 steps.
>
> **Comment:** How do the authors evaluate the robustness of the framework, separately in terms of hyperparameters during training, and sampling?
> **Response:** The only coupling between training and sampling is the precision schedule, as the encoding distribution should match the distribution of the latent variables during sampling as closely as possible. We included new ablations of the precision schedule, including an ablation of the final precision and results for a linear schedule similar to BFNs (see Section 4.3, Table 5, Figure 7). The takeaway is that as long as the final precision is chosen appropriately, the results only vary slightly. We provide clear instructions on how to find that final precision: It should be just high enough so that the reconstructor can predict the clean train samples flawlessly (see Figure 7).
> Except for the precision schedule, the sampler can be freely tuned without retraining. This includes varying the noise parameter $\gamma$, different discretization schemes, non-uniform time-stepping, and even completely different samplers like the FlowBack [2] sampler.  We find that the noise level $\gamma$ is by far the most significant hyperparameter to fine-tune (see the updated Section 4.3).
>
> We thank the reviewer again for their feedback and remain open to addressing any remaining or upcoming concerns.
>
> [2] **Song, Y., Shi, J., Gong, J., Xu, M., Ermon, S., Zhou, H.; Ma, W.. (2025).** Smooth Interpolation for Improved Discrete Graph Generative Models. Proceedings of the 42nd International Conference on Machine Learning

---

> > ### Comment · Reviewer_skZ3 · 2025-11-27
> >
> > Thank you for the reply.
> > I agree with other reviewers that adding clearer comparisons against continuous-space baselines will further strengthen the motivation and positioning.
> > I increase the score to 6.

---

> > > ### Author Response · Authors · 2025-11-27
> > >
> > > We thank the reviewer for their positive feedback and for raising the score.
> > >
> > > We are currently conducting additional ablations with GraphBFN to further isolate the effects of our modifications. Furthermore, we are implementing an experiment comparing with Fisher flow matching. We will post these results here if they are completed in time. If not, we will include them in the final version.

---

> > > ### Author Response · Authors · 2025-12-03
> > >
> > > We thank the reviewer again for their valuable feedback. As indicated in the global comment, we added two more comparisons against continuous-space baselines to strengthen the motivation and positioning:
> > >
> > > 1.⁠ ⁠**Comparison with GraphBFN**: We trained a GraphBSI model with a smaller initial variance, isolating the effect of sampling the belief at $t=0$ instead of taking a fixed value like with BFNs. Table 5 shows that for both values of $\beta_0$, the OU sampler outperforms the Flowback sampler on most metrics. Surprisingly, the performance of the Flowback sampler drops significantly when $\beta_0$ is increased, while a higher value of $\beta_0$ improves performance for the OU sampler.
> > >
> > > 2.⁠ ⁠⁠**Comparison with Fisher Flow Matching**: We trained a GraphBSI model on the synthetic benchmark proposed in [1] to compare with Fisher Flow Matching [1] and Dirichlet Flow Matching [2] (see Fig. 9). Our results illustrate that GraphBSI outperforms Dirichlet Flow Matching consistently and is competitive with Fisher Flow Matching.
> > >
> > > We again thank the reviewers for their valuable feedback.
> > >
> > > [1] *Davis, O., Kessler, S., Petrache, M., Ceylan, İ. İ., Bronstein, M., & Bose, A. J.* (2024). Fisher flow matching for generative modeling over discrete data. Advances in Neural Information Processing Systems, 37, 139054-139084.
> > >
> > > [2] *Stark, H., Jing, B., Wang, C., Corso, G., Berger, B., Barzilay, R., & Jaakkola, T.* (2024). Dirichlet flow matching with applications to DNA sequence design. arXiv preprint arXiv:2402.05841.

---

### Official Review · Reviewer_3cUV · 2025-11-04

**Soundness:** 3
**Presentation:** 3
**Contribution:** 3
**Rating:** 6
**Confidence:** 3

**Summary:**

This paper presents GraphBSI, a new one-shot generative model for discrete graphs based on Bayesian Sample Inference (BSI).
Unlike conventional diffusion or Bayesian Flow Network (BFN) models, GraphBSI performs generation by refining a belief distribution in parameter space rather than evolving discrete samples directly. The authors derive a categorical BSI formulation and show that in the continuous-time limit, it becomes a stochastic differential equation (SDE). They further generalize this to a family of SDEs with a noise control parameter γ, allowing smooth interpolation between deterministic probability-flow ODEs and stochastic samplers. On standard molecule generation benchmarks (GuacaMol and MOSES), GraphBSI achieves  superior results across most metrics.

**Strengths:**

Extends the Bayesian Sample Inference framework to discrete graphs.

Introduces both Euler–Maruyama and Ornstein–Uhlenbeck discretization schemes.

Outperforms baselines on key metrics with fewer sample steps.

**Weaknesses:**

How sensitive is GraphBSI to the choice of the precision schedule β(t)? The paper only mentioned a monotonically increasing schedule.

Since GraphBSI is similar to BFN, a direct comparison with GraphBFN regarding training time would better highlight its practical advantages.

In addition, GraphBFN is not included in Table 2. Including GraphBFN results would allow for a clearer assessment of GraphBSI’s improvements over prior BFN-based approaches. The sampling_step=50 for GraphBSI is also missed in Table 2.

**Questions:**

See Weaknesses.

---

> ### Author Response · Authors · 2025-11-24
>
> We thank the reviewer for their valuable feedback. In the following, we address their questions.
>
> **Comment:** How sensitive is GraphBSI to the choice of the precision schedule $\beta(t)$? The paper only mentioned a monotonically increasing schedule.
>
> **Response:** In our initial exploration, we tested different schedules but did not observe substantial differences in their performance. We added results trained with a linear schedule in the extended results table (Table 5) in the appendix.
>
> The only sensitivity we observed is to the final $\beta_\text{end}$ (see Table 7). If $\beta_\text{end}$ is too small, the graph is not completely denoised, and many random flips occur. On the other hand, if the value is too large, many function evaluations are used on an already denoised graph. We added a figure containing the results for different choices of $\beta_\text{end}$ in Figure 7.
>
> **Comment:** A direct comparison with GraphBFN regarding training time would better highlight its practical advantages.
> **Response:** Unfortunately, we are not able to provide a direct training runtime comparison as neither the code of GraphBFN nor their training details, i.e., number of epochs and training iterations, are available (the GitHub repository linked in their paper is empty as of 23rd November 2025).
>
> However, in terms of algorithmic computations, their training runtime should be equivalent to ours, as both methods optimize an $\ell_2$ loss. To still enable a fair comparison of GraphBSI and GraphBFN, we implement their sampling techniques with the same checkpoints as in our final evaluation. We included the results in Table 5.
>
> **Comment:** GraphBFN is not included in Table 2. Including GraphBFN results would allow for a clearer assessment of GraphBSI’s improvements over prior BFN-based approaches. The sampling_step=50 for GraphBSI is also missing in Table 2.
>
> **Response:** We thank the reviewer for noticing the missing rows. We included the results from GraphBFN and GraphBSI with 50 steps in the updated manuscript.
>
> We are happy to address any remaining or upcoming concerns.

---

### Official Review · Reviewer_w6nS · 2025-11-10

**Soundness:** 3
**Presentation:** 4
**Contribution:** 4
**Rating:** 8
**Confidence:** 4

**Summary:**

The paper extends Bayesian Sample Inference (BSI) to categorical/graph generation, introduces an SDE view with a controllable noise family ($\gamma$), and studies two discretizations (EM/OU). On molecular benchmarks, the method achieves competitive performance with relatively few function evaluations (e.g., 50/500 NFE). The contribution is integration-oriented: it adapts BFN/BSI ideas to discrete graphs with a clearer Bayesian update and an SDE formulation, offering a tunable trade-off between speed and quality. The writing is generally clear, but several theoretical statements and implementation details would benefit from refinement.

**Strengths:**

- **Originality**: Systematically ports BSI to **discrete/categorical graphs** and unifies deterministic ODE and stochastic sampling paths via **SDE + controllable noise**. This “bridging” removes some practical limitations of prior BFN-style derivations and constitutes originality via domain adaptation and simplification.
- **Quality**: A clear training objective (ELBO) and a family of samplers (controlled by $\gamma$) are provided, with strong **inference efficiency** (competitive at low NFE). Ablations on $\gamma$ and time grids are helpful; engineering appears solid.
- **Clarity**: Key intuitions (“belief contraction,” “same marginal family”) are conveyed through diagrams and concise algorithms; the separation of main text and appendix reads well.
- **Significance**: For **discrete structure generation** (notably molecular graphs), the **low-step, efficient** generation is practically relevant. Methodologically, it offers a clean, implementable link between BFN/BSI and diffusion-SDE paradigms, with potential impact on broader discrete domains.

**Weaknesses:**

1) **ELBO Reconstruction Term** In the main theorem, the reconstruction term is written as $\mathbb{E}[p(x|z)]$ instead of the standard $\mathbb{E}[\log p(x|z)]$. While the conclusion that this term is independent of $\theta$ still holds, the ELBO should use the expected log-likelihood. Please unify the notation in the main text and appendix.

2) **Weighting in Theorem 5** To be strictly consistent with the loss in the main text, a factor $\tfrac{1}{2}$ is needed, i.e.,
$$
\lambda(t) = \frac{\beta'(t)}{2} \cdot \frac{(\beta(t) + \beta_0)^2}{\beta(t)^2}.
$$
The current statement may double the scaling. A short clarification aligning Theorem 5 with Eq. (5) would help.

3) **Over-Strong “Affine Equivalence” Wording** The appendix claims an affine equivalence between the receiver distribution and BSI observations, but the covariance scaling does not match exactly. A safer phrasing is “same-order approximation in the small- $\alpha$ limit,” avoiding over-commitment.

4) **Final Decoding Inconsistency** The text mentions sampling from $\mathrm{Cat}(\mathrm{softmax}(z))$ while algorithms return `Quantize(f_\theta(\cdot,1))`. Since sampling vs. quantization affects diversity and faithfulness differently, please unify the terminology and state the default choice with a brief rationale.

5) **Empirical Support for “Same Marginal” is Indirect** The claim that different $\gamma$ share the same marginal is mainly supported through downstream metrics. A lightweight check in $z$ -space (e.g., comparing first and second moments at a few $t$, $\gamma$) would make the claim more tangible—no large-scale new experiments needed.

6) **Comparative/Robustness Details Could Be Lightly Strengthened** If space permits, adding small variance bands over a few seeds on key plots or clarifying a “same wall-clock budget” comparison protocol would improve perceived fairness and reproducibility.

**Questions:**

1) Could you add a minimal appendix probe comparing means/covariances of $z_t$ across a few $t$ and $\gamma$ values as an intuitive check for the “same marginal” claim? This can be very lightweight.
2) For Theorem 5, do you plan to include the missing $\tfrac{1}{2}$ factor in a revision/erratum and explicitly state the condition under which it is strictly aligned with Eq.(5)?
3) For the final decoding, which default path do you recommend—**categorical sampling** or **argmax quantization**—and why (stability, diversity, alignment with prior work)?
4) If adding figures is difficult, would you consider providing a reference configuration in the repo for a “**same wall-clock budget**” comparison (fixed batch/GPU/time) so others can verify fairness?
5) For the “affine equivalence” phrasing, would you consider noting in the appendix that “same-order approximation” is more accurate when covariance scalings differ, to avoid readers inferring strict equivalence?

---

> ### Author Response · Authors · 2025-11-23
>
> We thank the reviewer for their thorough feedback. In the following, we answer their questions.
>
> **Comment:** In the main theorem, the reconstruction term is missing a logarithm. While the conclusion that this term is independent of $\theta$ still holds, the ELBO should use the expected log-likelihood. Please unify the notation in the main text and appendix. **Comment:** To be strictly consistent with the loss in the main text, a factor $\frac{1}{2}$ is needed. The current statement may double the scaling. A short clarification aligning Theorem 5 with Eq. (5) would help. **Q:** For Theorem 5, do you plan to include the missing factor in a revision/erratum and explicitly state the condition under which it is strictly aligned with Eq.(5)?
>
> **Response:** We thank the reviewer for spotting the missing factor and the missing log. We have corrected the manuscript accordingly.
>
>
> **Comment:** The appendix claims an affine equivalence between the receiver distribution and BSI observations, but the covariance scaling does not match exactly. A safer phrasing is "same-order approximation in the small-$\alpha$  limit," avoiding over-commitment. **Q:** For the "affine equivalence" phrasing, would you consider noting in the appendix that "same-order approximation" is more accurate when covariance scalings differ, to avoid readers inferring strict equivalence?
>
> **Response**: Thank you for the suggestion. We updated the statement accordingly.
>
> **Comment:** Final Decoding Inconsistency The text mentions sampling from $\text{Cat(softmax)}(z)$ while algorithms return Quantize($f_\theta(\cdot,1))$. Since sampling vs. quantization affects diversity and faithfulness differently, please unify the terminology and state the default choice with a brief rationale. **Q:** For the final decoding, which default path do you recommend—categorical sampling or argmax quantization—and why (stability, diversity, alignment with prior work)?
> **Response:** Thank you for pointing this out. There is an interesting reason behind this discrepancy. Sampling from the final belief yields a simple ELBO since the final sampling step is completely independent of the reconstructor. However, we recommend returning a quantized reconstruction as it significantly improves sampling efficiency empirically. We added the explanation, including a plot that clarifies this, to Section 3.
>
> **Comment:** The claim that different $\gamma$ share the same marginal is mainly supported through downstream metrics. A lightweight check in  $z$-space (e.g., comparing first and second moments at a few $\gamma,t$ ) would make the claim more tangible—no large-scale new experiments needed. **Q:** Could you add a minimal appendix probe comparing means/covariances of $z_t$ across a few $\gamma$ and $t$ values as an intuitive check for the "same marginal" claim? This can be very lightweight.
> **Response:** We follow the reviewer's suggestion and added an example with a binary belief, showing the empirical quantiles at different points by the value of $t$ and $\gamma$ in Figure 6 in the appendix.
>
> **Comment:** If space permits, adding small variance bands over a few seeds on key plots or clarifying a "same wall-clock budget" comparison protocol would improve perceived fairness and reproducibility. **Q:** If adding figures is difficult, would you consider providing a reference configuration in the repo for a "same wall-clock budget" comparison (fixed batch/GPU/time) so others can verify fairness?
> **Response:** As BSI is a generalization of BFNs, we are able to train a BFN model with our codebase, enabling a fair comparison. We included results with the FlowBack sampler. All code will be published upon acceptance, including the configuration files used to reproduce our results.
>
> We thank the reviewer again and are happy to address any remaining or upcoming questions.

---

### Author Response · Authors · 2025-11-24

We thank the reviewers for their feedback. While we have addressed their concerns in detail under each review, we would like to summarize important changes in the manuscript.

- **Moses Evaluation**: We found that for the Moses benchmark, we evaluated FCD, SNN, and Scaf on the `test` split instead of the `test_scaffolds` split, which is the standard benchmark split. The updated results with the `test_scaffolds` split in Table 1 remain competitive with the strongest baselines in SNN and Scaf, and improve upon them in FCD.
- **DNA Sequence Generation:** We have included a new experiment evaluating our model on the FlyBrain dataset to compare with Dirichlet Flow Matching. The experiment is shown in Appendix B.
- **Connection to BFN and Diffusion models:** We have included a detailed theoretical comparison to diffusion models and extended our comparison to BFNs in Appendix A.

We again thank the reviewers for their valuable feedback.

---

### Author Response · Authors · 2025-12-03

Dear reviewers,
We conducted **two new experiments** to provide further comparisons to continuous baselines:

1.⁠ ⁠**Isolation of performance gains**: We trained a GraphBSI model with a smaller initial variance, isolating the effect of sampling the belief at $t=0$ instead of taking a fixed value like with BFNs. Table 5 shows that for both values of $\beta_0$, the OU sampler outperforms the Flowback sampler on most metrics. Surprisingly, the performance of the Flowback sampler drops significantly when $\beta_0$ is increased, while a higher value of $\beta_0$ improves performance for the OU sampler.

2.⁠ ⁠⁠**Comparison with Fisher Flow Matching**: We trained a GraphBSI model on the synthetic benchmark proposed in [1] to compare with Fisher Flow Matching [1] and Dirichlet Flow Matching [2] (see Fig. 9). Our results illustrate that GraphBSI outperforms Dirichlet Flow Matching consistently and is competitive with Fisher Flow Matching.

We again thank the reviewers for their valuable feedback.

[1] *Davis, O., Kessler, S., Petrache, M., Ceylan, İ. İ., Bronstein, M., & Bose, A. J.* (2024). Fisher flow matching for generative modeling over discrete data. Advances in Neural Information Processing Systems, 37, 139054-139084.

[2] *Stark, H., Jing, B., Wang, C., Corso, G., Berger, B., Barzilay, R., & Jaakkola, T.* (2024). Dirichlet flow matching with applications to dna sequence design. arXiv preprint arXiv:2402.05841.

---

### Meta-Review · Area_Chair_KSSv · 2026-01-05

**Summary:**

This paper proposes GraphBSI, a discrete Bayesian sample inference framework for graph generation that maintains and refines a belief over graphs in the continuous space of distribution parameters, and derives an SDE formulation with a controllable noise family together with practical samplers for efficient generation. Reviewer concerns about positioning and baselines were constructively addressed in discussion through clearer comparisons to continuous space alternatives and GraphBFN related baselines, and at least one reviewer explicitly indicated an increased score after these additions. I recommend acceptance.

**Reviewer Concerns:**

Addressed in rebuttal or discussion: clearer positioning against GraphBFN and additional continuous space baseline comparisons, plus added experimental evidence beyond the original graph benchmarks.
Still outstanding: sensitivity to hyperparameters and some remaining requests for broader ablations and tighter presentation of a few theoretical terms, though these do not appear to block acceptance.

**Reviewer Scores:**

w6nS stays at 8.
3cUV stays around 6.
skZ3 increased to 6 after baseline clarifications.
WtYP would likely increase from 4 to 6 given they stated their concerns were resolved and they were willing to raise the score.

---

### Decision · Program_Chairs · 2026-01-26

Accept (Poster)